# Validation of the rabbit pain behaviour scale (RPBS) to assess acute postoperative pain in rabbits (*Oryctolagus cuniculus*)

Renata Haddad Pinho[1]☺, Stelio Pacca Loureiro Luna[2]☺*, Pedro Henrique Esteves Trindade[2]☺, André Augusto Justo[3]☺, Daniela Santilli Cima[1]☺, Mariana Werneck Fonseca[1]☺, Bruno Watanabe Minto[4]☺, Fabiana Del Lama Rocha[4]☺, Amy Miller[5]‡, Paul Flecknell[6]‡, Matthew C. Leach[6]‡

1 Department of Surgical Specialties and Anesthesiology, Botucatu Medical School, São Paulo State University (Unesp), Botucatu, São Paulo, Brazil, 2 Department of Veterinary Surgery and Animal Reproduction, School of Veterinary Medicine and Animal Science, São Paulo State University (Unesp), Botucatu, São Paulo, Brazil, 3 Department of Surgery, School of Veterinary Medicine and Animal Science, University of São Paulo (USP), São Paulo, São Paulo, Brazil, 4 Department of Veterinary Clinics and Surgery, School of Agricultural and Veterinary Sciences, São Paulo State University (Unesp), Jaboticabal, São Paulo, Brazil, 5 School of Health and Life Sciences, Teesside University, Middlesbrough, United Kingdom, 6 School of Natural and Environmental Science, Newcastle University, Newcastle Upon Tyne, United Kingdom

☺ These authors contributed equally to this work.
‡ AM, PF and MCL also contributed equally to this work.
* stelio.pacca@unesp.br

**Data Availability Statement:** All relevant data are within the paper and its Supporting Information files.

## Abstract

Considering the widespread use of rabbits in research that potentially causes pain and discomfort and the limited number of pain assessment validated tools in this species, we aimed to develop and validate a scale of acute postoperative pain in rabbits (RPBS). Footage of 58 rabbits from previous studies were used, recorded at '*baseline*' (before orthopaedic and soft tissue surgeries), '*pain*' (after surgery), '*analgesia*' (after analgesic), and '*24h post*' (24 hours after surgery). The videos were randomised and assessed twice by four evaluators, within one-month interval between evaluations. After content validation, RBPS was further refined using the criteria from the validation. According to the principal component analysis, RPBS was considered unidimensional. The intra- and inter-observer reliability was excellent (ICC>0.80) for all evaluators. There was a high Spearman's correlation of the RPBS with unidimensional scales (>0.80) and a moderate correlation with the Rabbit Grimace Scale (0.68), confirming criterion validity. According to the mixed linear model, the scale was responsive, shown by the increase in pain scores after surgery. Construct validity was confirmed by known-group approach and internal relationships among items. Adequate item-total correlation (>0.3) was observed for all items, except for the attention to the affected area (0.04). The internal consistency was very good (Cronbach's α coefficient = 0.78; Mcdonald's ω coefficient = 0.83). The cut-off score for rescue analgesia was ≥3, with an area under the curve >0.95, demonstrating a high discriminatory capacity of the instrument. Scores 3 and 4 were within the uncertainty diagnostic zone. Specificity was 87% and sensitivity was 90%. It was concluded that the RPBS presented content, criterion, and construct validities, responsiveness, and reliability to assess acute pain in rabbits submitted to

**Funding:** RHP - sponsored by FAPESP (São Paulo Research Foundation) - M.Sc. grant 2018/17839-7 SPLL- Financial support by FAPESP (São Paulo Research Foundation) – thematic project 2017/12815-0 (http://www.fapesp.br/en/) This study used videos from previous funded research; Miller et al. 2021 was funded by the National Centre for 3Rs (NC3Rs Grant/Award No. G1100563), Leach et al 2009, was funded by Boehringer Ingelheim Vetmedica GmbH, Germany. The funders had no role in study design, data collection and analysis, decision to publish, or preparation of the manuscript.

**Competing interests:** The authors have declared that no competing interests exist.

orthopaedic and soft tissue surgeries. The cut-off for rescue analgesia serves as a basis for the administration of analgesics to rabbits submitted to painful procedures.

## Introduction

The principles of the 3Rs (Replacement, Reduction and Refinement) guide global legislation regarding the use of animals in experiments. Specifically, the principle of refinement contemplates the reduction in stress and suffering of animals, which promotes the use of protocols that prevent and treat pain [1]. However, pain treatment after experimental surgery is still neglected in rodents and rabbits [2]. Pain is likely underestimated and undertreated in this species due to our inability to recognize the phenomen [3, 4].

Rabbits are widely used as experimental models, often being subjected to procedures that potentially cause pain [5]. In parallel to their use in research, rabbits are becoming increasingly popular as pets [6]. Under these circumstances they may experience painful conditions, like in dental disease, trauma, fractures, and sterilization surgeries [7].

Behaviour is a non-invasive method used to recognize pain in laboratory animals [8–13]. However, to effectively recognize pain, behavioural analysis must be based on validated scales [14] with defined cut-off points for analgesic administration [15].

Based on scales developed for rats and mice [16, 17], the Rabbit Grimace Scale (RbtGS) was developed and partially validated to assess pain in rabbits [18]. The tool was developed through images photographed while performing an identification tattoo on the ears. However, RbtGS has not yet been validated for assessing postoperative pain. Recent research associated this tool with physiological and behavioural parameters to assess clinical pain in pet rabbits, developing the composite pain scale for assessing and quantifying pain in rabbits (CANCRs) [19]. Recently, a study developed in parallel with the present study established the Bristol Rabbit Pain Scale (BRPS), based on the selection of behavioural descriptors approved in five phases by experts in the field and on evaluation of videos [20]. However, none of the tools described above to assess pain in rabbits has determined a cut-off point for rescue analgesic. It is required that the validated instrument not only recognizes pain, but presents a representative score in decision-making for analgesic intervention.

In other species, behavioural scales designed to assess pain stand out for providing evidence of validity, reliability, and sensitivity [21] and because they have defined cut-off points, such as that developed for cats [15], cattle [22], pigs [23], sheep [24], and donkeys [25].

Based on the methodology used to validate pain scales in the aforementioned species [15, 22–25], the current study aimed to validate a scale to assess acute pain resulting from different types of surgeries in rabbits.

## Materials and methods

The study covers two of the principles of the 3 Rs (*Reduction and Refinement*) [1]. This is a bicentric, opportunistic, randomized, and blinded study that aims to promote animal welfare by providing a tool to assess pain and facilitate decision-making for its treatment *(Refinement)*. Footage taken during the development of three other studies of the perioperative periods of 58 rabbits, submitted to three types of surgeries, was used *(Reduction)* (Tables 1 and 2).

This study was designed according to the Consensus Based Standards for the Selection of Health Measurement Instrument (COSMIN) checklist and terminology for assessing the methodological quality of studies [26, 27].

**Table 1. Description and summary of the five experimental groups based on the three previous studies.**

| Surgery/ reference | Group | n | Premedication | Anaesthetic Induction | Anaesthetic Maintenance | Postoperative rescue analgesic | Type of accommodation, food and water | Environmental enrichment | Duration of original footage |
|---|---|---|---|---|---|---|---|---|---|
| **Orchiectomy** [28] | ORC-Melox | 8 | 0.2mg/kg meloxicam (SC) | 10mg/kg propofol (IV) | Sevoflurane (4–6%) | Not required: 0.05mg buprenorphine (SC) | Stalls with wood shavings (3cm) | Wood shavings, cardboard box, cardboard tube, cat litter tray and chew blocks | 15 minutes |
| | ORC-Multi | 8 | 0.6mg/kg meloxicam + 0.03mg/kg buprenorphine (SC), local infiltration with 4mg/kg lidocaine and 2mg/kg bupivacaine | | | | | | |
| **Ovariohysterectomy*** [11] | OVH-Pla | 7 | placebo (Oral) | | Sevoflurane | | Stalls with wood shavings (3cm). Dry feed and water available *ad libitum* | Cardboard roll, wood shavings | 20 minutes |
| | OVH-Melox | 7 | 1mg/kg meloxicam (Oral) | | | | | | |
| **Partial radial ostectomy** [12] | Ortho | 28 (11 females and 17 males) | 5mg/kg pethidine (IM) | Isoflurane | Isoflurane + 2µg/kg fentanyl (IV) | 2mg/kg morphine + 1mg/kg meloxicam (IM) | 60cm$^3$ cage, with bars. Dry feed and water available *ad libitum* | Eucalyptus pine cones replaced frequently and carrots supplied before the start of each film recording | Five minutes |

* The experimental groups OVH-Pla and OVH-Melox refer to *groups 1* (Placebo) and *4* (1mg/kg meloxicam) and the videos of the *baseline* time point for these groups correspond to those filmed in the morning (*first period—morning*) [11]

** The Ortho group refers to the filming of animals in the absence of the observer (*Ab*) [12]. IM: intramuscular; Oral: orally; SC: subcutaneous; IV: intravenous.

## Experimental groups

**Orchiectomy-Melox (ORC-Melox) and Orchiectomy-Multimodal (ORC-Multi).** The procedures in these groups were performed at Newcastle University, conducted in accordance with the Animals (Scientific Procedures) Act 1986, and European Directive EU 2010/63, with the approval of the Newcastle University Animal Welfare and Ethics Body. These studies employed a strict 'rescue' analgesia policy. If any animal was deemed to be in greater than mild pain (i.e., if rabbits displayed > 4 validated pain behaviours within 5 minutes) assessed by an independent veterinarian, then buprenorphine (0.05 mg/kg SC) was immediately administered, and the animal was removed from the study. No animals were deemed to require intervention analgesia. All animals recovered from surgery uneventfully.

On the day of surgery, the 16 rabbits were transported to the theatre preparation room using a pet carrier. All rabbits were administered their allocated dose of meloxicam

**Table 2. Filming time points evaluated in the five groups of rabbits undergoing surgery.**

| Group | Baseline | Pain | Analgesia | 24h post |
|---|---|---|---|---|
| **ORC-Melox** | 46 hours before surgery | 1 hour after surgery | Not evaluated | 24 hours after surgery |
| **ORC-Multi** | | | | |
| **OVH-Pla** | 22 hours before surgery (morning) | 3 hours after surgery | | 27 hours after surgery |
| **OVH-Melox** | | | | |
| **Ortho** | 24 hours before surgery | 1 hour after anaesthetic recovery | 3 hours after analgesic rescue | 24 hours after anaesthetic recovery |

(ORC-Melox–n = 8) or meloxicam and buprenorphine (ORC-Multi–n = 8), 30 minutes prior to anaesthesia. Anaesthesia was induced by intravenous propofol (10 mg/kg), beginning between 9am and 10:30am. Rabbits were then placed on a heating blanket (Harvard Apparatus, Edenbridge, UK), intubated, and the anaesthesia was maintained with sevoflurane (4–6%) in oxygen (4l/min). Rabbits were placed in dorsal recumbency and the scrotal area was shaved. The scrotum was sprayed with chlorohexidine (Hydrex Dema Spray, Adam Healthcare, Leeds, UK) and skin infiltrated with local anaesthetic (ORC-Multi) or saline (ORC-Melox), followed by castration surgery using a 2-cm incision, observing full aseptic technique. A 2-cm incision allowed easy visualization of the testes. Testes were blunt dissected, the cord infiltrated with local anaesthetic (ORC-Multi) or saline (ORC-Melox). The testis was clamped above infiltration and transfixed and ligated with 3.0 Vicryl (round-bodied needle). Splash block area and SC infiltration with local anaesthetic (ORC-Multi) or saline (ORC-Melox) was performed. Once the castration was complete, the dead space was closed with 3.0 Vicryl (round-bodied needle), and subcuticular skin closure was performed with Vicryl and a cutting edge needle (3.0 Vicryl).

The same experienced surgeon carried out all procedures. Following surgery, rabbits recovered in an incubator (25˚C) for 1 hour where they were closely monitored by animal care staff. The rabbits were then transferred back to the filming pen for the first post-surgery video recording. Daily wound checks and health monitoring were carried out until the wound was fully healed. Filming was carried out remotely (Table 2).

**OVH-Placebo (OVH-Pla) and OVH-Meloxicam (OVH-Melox).** Ethical issues were the same as applied for the ORC-Melox and ORC-Multi groups.

The filming of the OVH-Pla and OVH-Melox groups was carried out in the absence of an observer [11] and details of methodology including drugs, doses, and evaluated time points can be found in Tables 1 and 2.

Further details regarding the management, anaesthetic, and surgical procedures of animals in the OVH-Pla and OVH-Melox groups can be found in a previous study [11].

**Orthopaedic surgery (Ortho).** The study was approved by the Ethical Committee for the Use of Animals in Research, of the School of Veterinary Medicine and Animal Science and School of Agricultural and Veterinary Sciences, São Paulo State University (Unesp), under protocol numbers 0156/2018 and 019155/17, respectively. The study follows the Brazilian Federal legislation of CONCEA (National Council for the Control of Animal Experimentation).

The footage of the Ortho group in the absence of an observer used in this study corresponds to the perioperative periods (Table 2) of 28 rabbits (11 females and 17 males) undergoing orthopaedic surgery for another parallel study [12]. We chose to use the videos in the absence of the observer to standardize with the other groups described above. Details of drugs, accommodation, and filming time points are described in Tables 1 and 2 and in a previous article [12]. Before the onset of recordings, a piece of carrot, and a new Eucalyptus pinecone were offered as motivational items.

In addition to the perioperative time points described in Table 2, eight animals from the Ortho group were filmed 24 hours before surgery at 8am, 2pm, and 8pm, to assess possible behavioural changes according to the period of the day.

**Elaboration of the pain scale.** The scale was based on behaviours identified in previous studies, considered important to assess pain in rabbits [11, 12, 18, 29–34].

**Evaluation of the videos.** The videos of the 58 animals were edited by the main researcher of the study to generate 2-to-3-minute videos that proportionally represented the duration and frequency of the behaviours observed in the original videos. This edition was based on the ethogram recorded in the three previous studies [11, 12, 28]. For example, if the rabbit was lying down for 5 minutes during the 15-minute original footage (1/3 of the time), edits were

performed to guarantee that the rabbit was lying down for 1 minute of the 3-minute video clip (1/3 of time).

In total: I) 112 (Ortho) and 90 videos (ORC and OVH) were produced, corresponding to *baseline*, *pain* (time point of greatest expected pain), *analgesia* (after rescue analgesic; only in the Ortho group), and *24h post* and II) 24 videos from the eight animals in the Ortho group in pain-free conditions one day before surgery (8am, 2pm, and 8pm).

**Selection and training of evaluators.** All evaluators were veterinarians with two-year residency in veterinary anaesthesiology and around 4 years of experience in the area (RHP, AAJ, DSC, and MWF)

For training, evaluators watched videos of each behaviour on the scale for familiarization and received instructions on how to fill out the scales. One week after training, the observers evaluated ten videos. After a further one-week break, the observers rated the same ten re-randomized videos and gave them a second score. This assessment served as a test to ensure that the intra-observer reliability (ICC) was above 80% for the sum of the scale. The evaluators were then considered able to start the definitive evaluations. For the RbtGS, the evaluators received images and descriptors of the FAUs before starting the evaluations.

**Video evaluation.** The evaluators watched the videos of each rabbit in random order blinded to treatment and time points. The evaluators only knew which surgery the rabbit was or would be submitted to identify the region of the affected area. The evaluation period lasted five weeks. The evaluators watched the videos as many times as necessary to score them.

After watching each video, based on their clinical experience, the evaluators indicated whether they would provide rescue analgesic in that situation or not (RA– 0 or 1). Next, they evaluated pain using the numerical scale (NS—0 to 10), the simple descriptive scale (SDS—1 to 4), the visual analog scale [VAS—0 (no pain) to 100 mm (worst possible pain)], the proposed scale (RPBS—0 to 12), and the facial scale (RbtGS—0 to 2) [18]. The evaluators identified which sub-item(s) of the scale gave rise to the score of each item. To score the facial scale, the evaluators were instructed to score the highest scores for each item observed on the video.

One month after completing the first evaluation, the evaluators watched and scored the same re-randomized videos again, to establish intra-observer reliability.

## Data and statistical analysis

Statistical analyses for scale refinement and validation (Table 3) were performed in R software in the Rstudio integrated development environment [35]. For all analyses, an α of 5% was considered. The validation analyses were performed using the scores given at all time points by all evaluators grouped in phases 1 and 2.

## Results

The scale was divided into the following behavioural categories: posture, activity, interaction and appetite, facial expression, attention to the affected area, and miscellaneous behaviours (S1 Table).

## Refinement of the RPBS

All items were approved by the expert committee during the content validation (S2 Table), and thus included in the first version of the scale (pre-refinement) evaluated by the observers (S1 Table).

To refine the scale (S3 Table), items approved in at least seven of the following criteria were selected: content validation, frequency of occurrence in the '*pain*' time point, principal component analysis, intra- and inter-observer reliability, responsiveness, item-total correlation,

**Table 3. Statistical analysis for refinement (R) and validation (V) of the RPBS [24].**

| Statistical analysis | Description | Statistical test |
|---|---|---|
| **Content validation**[R] | The following steps were performed: 1) a list of pain-related behaviours reported in the previous ethograms and 2) a committee composed of three veterinarians experienced in assessing pain (two senior veterinary anaesthetists experienced in animal pain assessment and a laboratory animal veterinarian) who did not participate in the evaluations, sorted each sub-item within each item of the scale into relevant (+1), do not know (0), or irrelevant (-1). | All the values of each sub-item (-1, 0, or 1) were added and the total was divided by the number of observers. Items with a total score > 0.5 were included in the scale [36]. |
| **Occurrence of the items and sub-items**[R] | The percentage of occurrence of each item and sub-item of the scale was evaluated. | Descriptive analysis. Interpretation: items and sub-items with occurrence > 15% of the total number of rabbits at the *pain* time point were accepted after refinement [24]. |
| **Distribution of scores**[V] | Distribution of the frequency of the presence of the scores 0, 1, and 2 of each item at each time point for each group. | Descriptive analysis. |
| **Multiple association**[RV] | Multiple association between items was analysed at all time points grouped (MG) using principal component analysis, to define the number of dimensions determined by different variables that establish the scale extension. Confirmatory factor analysis was used to compare one dimension, two uncorrelated dimensions and two correlated dimension models [37] | Principal component analysis ("princomp" and "get_pca_var" functions from the "stats" and "factoextra" packages respectively). According to the Kaiser criterion [38], representative dimensions of the components were selected with eigenvalue > 1 and variance > 20. Horn's Parallel Analysis [39] ("fa.parallel" function from the "psych" package), Marchenko-Pastur limit ("chooseMarchenkoPastur" function from the "PCAtools" package), and Gavish-Donoho method [40, 41] ("chooseGavishDonoho" function from the "PCAtools" package), were performed to determine the optimal number of dimensions to be retained. For the biplot, confidence ellipses were produced with significant levels of 95% to show the density of scores at each time point or each group. Loading value $\geq 0.50$ or $\leq -0.50$ were considered for significant association. Confirmatory factor analysis ("cfa" function from the "lavaan" package") was used. The parameters chi-square, comparative fit index, Tucker Lewis index, log likelihood, root mean square error of approximation, Akaike and Bayesian information criterion were used to compare a one dimension, two uncorrelated dimensions and two correlated dimension models. |
| **Intra-observer reliability**[RV] | The level of agreement of each observer with themselves was estimated by comparing the two phases of assessment, using the scores of each item, the total sum of the RPBS, NS, SDS, VAS, and the need for rescue analgesia. | For the scores of the items of the RPBS, the NS, SDS, and the need for rescue analgesia, the weighted kappa coefficient (kw) was used; the disagreements were weighted according to their distance to the square of perfect agreement. The 95% confidence interval (CI) kw ("cohen.kappa" function of the "psych" package) was estimated. For the VAS, the intraclass correlation coefficient (ICC) type "agreement" was used and its 95% CI ("icc" function of the "irr" package) [42–44]. For the sum of the RPBS, the consistency type ICC and its 95% CI was calculated based on single measures. Interpretation of kw and ICC: very good 0.81–1.0; good 0.61–0.80; moderate 0.41–0.60; reasonable 0.21–0.4; and poor < 0.2 [45]. The kw and ICC > 0.50 were used as criteria to refine the scale. |
| **Inter- observer reliability**[RV] | A matrix was generated to assess the level of agreement among all observers, using the scores for each item, the total sum of the RPBS, NS, SDS, VAS, and the need for rescue analgesia | |
| **Criterion validity**[RV] | 1) Concurrent criterion validity (comparison with a validated instrument)—the correlation of the sum of the RPBS was estimated with the NS, SDS, VAS, and RbtGS. For the RbtGS, the average of their sum was used, disregarding the items that were not subjected to evaluation according to the formula $$RbtGs = \frac{sum\ of\ FAU\ scores\ assessed}{number\ of\ FAUs\ assessed}$$ FAU = facial action unit | Spearman rank correlation coefficient (rs; "rcorr" function of the "Hmisc" package). Interpretation of the degree of correlation rs (p < 0.05): 0–0.35 low correlation; 0.35–0.7 moderate correlation; 0.7–1.0 high correlation [46] |
| | 2) Concurrent criterion validity—the agreement between each observer vs all other observers (reproducibility). | See the description above for inter-observer reliability. |
| | 3) Predictive criterion validity—was assessed by the number of rabbits that should receive rescue analgesia according to the Youden index (described below) in the time point of greatest pain (*pain*). | Descriptive analysis. |

(*Continued*)

**Table 3.** (Continued)

| Statistical analysis | Description | Statistical test |
|---|---|---|
| **Responsiveness**[RV] | Responsiveness–the scores of each item and the total score of the RPBS, NS, SDS, VAS, and the need for rescue analgesia over time were compared for each group (OVH-Pla, OVH-Melox, ORC-Melox, ORC-Multi, and Ortho). | For each item of the scale, the Friedman's test was used, followed by Dunn's multiple comparison post-hoc test. |
| | | For the dichotomous variable, need for rescue analgesic, logistic regression analysis ("glm" function of the "stats" package) was applied using the post hoc Tukey test ("lsmeans" function of the "lsmeans" package). The normality of the model residuals ("residuals" function of the "stats" package) for the dependent variable (RPBS) showed Gaussian distribution for groupings according to the quantile-quantile and histogram graphs ("qqnorm" and "histogram" functions of the "stats" and "lattice" packages, respectively), thus, mixed linear models ("lme" function of the "nlme" package) were applied. The residual distribution was not considered normal for other dependent variables (RPBS separated by group) and, therefore, generalized mixed linear models ("glmer" function of the "lme4" package) were applied. For all models, time points, observers, groups (except when the groups were grouped) and phases were included as fixed effects and the individuals as random effect. Differences over time and intergroups were performed by the Bonferroni test as a post hoc test [24]. |
| **Construct Validity**[RV] | Construct validity was tested by four approaches [26, 27, 47]: | 1. Please see Responsiveness |
| | 1. Three-hypothesis test: 1) if the scale really measures pain, the score after surgery (*pain*) should be higher than the preoperative score (*baseline* < *pain*), 2) the score should decrease after analgesia (*pain* > *analgesia*), and 3) over time (*pain* > *24h post*) | 2.2. For the comparisons between negative control Ortho-group rabbits and Ortho-group rabbits suffering pain, Mann-Whitney test was performed. |
| | 2. Known-group validity. | 2.2. For comparisons between groups (OVH-Pla, OVH-Melox, ORC-Melox, ORC-Multi, and Ortho), mixed linear model was performed as explained before for responsiveness. |
| | 2.1. The scores of the *pain* time point recorded in the morning in the Ortho-group rabbits [before 12pm (n = 16)] were compared with the scores of the pain-free negative control rabbits filmed in the morning (8am) (n = 8). The same comparison was performed for the scores of the *Pain* time point recorded after 12pm (n = 12) versus the scores of pain-free rabbits recorded in the afternoon (2pm; n = 8). In addition, the scores of the pain-free negative control rabbits (8am + 2pm; n = 16) were compared with all rabbits in the Ortho group at the *Pain* time point (n = 28). | |
| | 2.2. Comparison of the pain scores between groups (OVH-Pla, OVH-Melox, ORC-Melox, ORC-Multi, and Ortho). | |
| | 3. Internal relationships among items according to all criteria used in statistical analysis (principal component analysis, internal consistency and item-total correlation) | |
| | 4. Relationships with the scores of other instruments, as described for criterion validity | |
| **Behaviour changes according to the time of the day**[V] | The scores of each item and the sum of the RPBS at 8am, 2pm, and 8pm in pain-free conditions (before surgery) were compared in eight animals from the Ortho group. | See construct validity |
| | | Expected interpretation: morning = afternoon = night |
| **Item-total correlation**[RV] | The correlation of each item with the total score, excluding the evaluated item, was estimated to analyse homogeneity, the inflationary items, and the relevance of each item of the scale. | Spearman rank correlation coefficient (r; "rcorr" function of the "Hmisc" package). Interpretation of correlation r: suitable values 0.3–0.7 [48]. Items were accepted if $r_s$ > 0.3. |
| **Internal consistency**[RV] | The consistency (interrelation) of the scores of each item on the scale was estimated. | Cronbach's alpha coefficient (α; "cronbach" function of the "psy" package) and McDonald's omega coefficient [49] were performed based on results from multiple association (ω; "omega" function of the "psych" package). Cronbach's alpha coefficient interpretation: 0.60–0.64, minimally acceptable; 0.65–0.69 acceptable; 0.70–0.74 good; 0.75–0.80 very good; and >0.80 excellent [50]. McDonald's omega coefficient interpretation: 0.65–0.80, acceptable; >0.80 strong reliability evidence [49]. |

*(Continued)*

**Table 3.** (Continued)

| Statistical analysis | Description | Statistical test |
|---|---|---|
| **Specificity and Sensitivity**[RV] | The scores of RPBS at *baseline* were transformed into dichotomous ("0"—absence of pain expression behaviour for a given item; "1" and "2"—presence of pain expression behaviour). | $Sp(baseline) = \frac{TN}{TN+FP}$ |
| | | $Sp$ = specificity. TN = true negative (scores that represented painless behaviours—"0"—at the time when the animals were expected to have no pain, since it was before surgery—*baseline*). FP = false positive (scores that represented pain expression behaviours—1 or 2—at the time when the animals were expected to have no pain, since it was before surgery–*baseline*). Interpretation: excellent 95–100%; good 85–94.9%; moderate 70–84.9%; not sensitive <70%. Only items $\geq$ 70% were included after refinement. |
| | | $S(pain) = \frac{TP}{TP+FN}$ |
| | | $S$ = sensitivity. TP = true positive (scores that represented pain expression behaviours—1 or 2—at the time the animals were expected to have pain, since it was after surgery—*Pain*). FN = false negative (scores representing painless behaviours—0—at the time the animals were expected to have pain, since it was after surgery—*Pain*). Interpretation: excellent 95–100%; good 85–94.9%; moderate 70–84.9%; not sensitive <70%. Only items $\geq$ 70% were included after refinement. |
| **Rescue analgesic point**[V] | The need for analgesia according to the clinical experience, after the observers had watched the videos, was used as the true value and the total score of the RPBS, NS, SDS, and VAS as a predictive value to build a ROC curve. The cut-off point for rescue analgesia was determined based on the Youden index and its diagnostic uncertainty zone. The AUC was calculated and indicate the discriminatory capacity of the test. | $YI = (S + Sp) - 1;$ |
| | | $YI$ = Youden Index; $S$ = sensitivity; $Sp$ = specificity. Analysis of the receiver operating characteristic curve (ROC; "roc" function of the "pROC" package) by a non-parametric approach [51] and the area under the curve (AUC): graphical representation of the relationship between the $S$ and 1-$Sp$. $YI$ is the point of greatest sensitivity and specificity simultaneously, determined by the ROC curve [15, 52]. Interpretation: AUC $\geq$ 0.90 indicates high discriminatory capacity of the scale [53]. |
| | | The diagnostic uncertainty zone was determined by two methods, calculating: 1[st]) the 95% confidence interval (CI) replicating the original ROC curve 1,001 times by the bootstrap method ("ci. coords" and "ci.auc" functions of "pROC" package) and 2) the interval between the sensitivity and specificity values of 0.90. The highest interval of these two methods was considered the diagnostic uncertainty zone, which indicates the diagnostic accuracy [54, 55] |
| | The frequency and percentage of animals scored in the diagnostic uncertainty zone of the cut-off point were calculated. | Descriptive statistical analysis |

Adapted from [24]. Scales: numerical (NS), simple descriptive (SDS), visual analogue (VAS), Rabbit Grimace Scale (RbtGS), facial action unit (FAU).

The validation analyses were performed using the scores given at all time points by all evaluators grouped in phases 1 and 2. For all analyses, an α of 5% was considered.

MG—data of grouped time points (*baseline + pain + analgesia + 24h post*).

internal consistency, sensitivity, and specificity (Table 3). For the sub-items, those approved in at least three of the content validation criteria, frequency of occurrence at the *'pain'* time point, intra- and inter-observer reliability, responsiveness, sensitivity, and specificity were selected. Only the sub-item "spasms" was excluded as it only met the content validity and specificity criteria.

The final post-refinement scale presented six items, with scores from 0 to 2 and a total sum from 0 to 12 (Table 4).

## Principal component analysis

All methods used to determine the optimal number of dimensions for retention indicated one dimension. The only exception was the Horn's Parallel Analysis which suggested two dimensions. The principal component analysis detected only one representative dimension with an eigenvalue > 1 and variance > 20 (Table 5 and Fig 1). All items except "attention to the

**Table 4. Final version of the behavioural scale to assess acute postoperative pain in rabbits (RPBS) after refinement.**

| RABBIT PAIN BEHAVIOUR SCALE (RPBS) | | |
|---|---|---|
| **Item** | **Video examples/ score** | |
| **1) Posture** | | |
| A) Moves around normally and/or jumps | S1 Video | https://youtu.be/5rFBxK2wXXk |
| B) Exhibits bipedal or quadrupedal position (with the four limbs extended vertically) | S2 Video | https://youtu.be/Q3DONFk5ydQ |
| | S3 Video | https://youtu.be/LgJOxzulkWY |
| C) Walks at a very slow pace | S4 Video | https://youtu.be/1-LEmiIKRmI |
| D) Lies for most of the time | S5 Video | https://youtu.be/uAw6x96VITU |
| E) Does not move for most of the observation time | S6 Video | https://youtu.be/ELl6yOpm7zs |
| Presence of state A and/or B only* | 0 | |
| Presence of one of states C, D, or E | 1 | |
| Presence of two or more of states C, D, or E | 2 | |
| *The score will only be 0 when there are no C, D or E behaviours | | |
| **2) Activity** | | |
| A) Moves normally and/or when stationary performs normal activity* | S7 Video | https://youtu.be/m1g24h-MnLw |
| B) Moves little and does not perform normal activity | S8 Video | https://youtu.be/ZVk0RdUDV8c |
| C) Is immobile and does not perform normal activity | S9 Video | https://youtu.be/4_X59ngfXQ4 |
| *Interacts with environmental enrichment objects (pine cone, toy and others), eats, drinks water, digs in shavings, exhibits self-cleaning behaviour, sniffs the environment | | |
| Presence of state A | 0 | |
| Presence of state B | 1 | |
| Presence of state C | 2 | |
| **3) Interaction and Appetite** | | |
| A) Interacts with environmental enrichment objects* | S10 Video | https://youtu.be/Sltyrw3zNFQ |
| | S11 Video | https://youtu.be/dxsNa1dadSo |
| B) Eats** | S12 Video | https://youtu.be/xogndZKVyHY |
| | S13 Video | https://youtu.be/CnGLfw9dlZk |
| C) Sniffs the environment | S14 Video | https://youtu.be/kD2Yeb9j-tc |
| D) Exhibits self-cleaning behaviour (grooming), with the exception of the affected area | S15 Video | https://youtu.be/H6E-7GfrpRk |
| | S16 Video | https://youtu.be/EjgKQIwhEkc |
| The rabbit presents more than one of these behaviours | 0 | |
| The rabbit presents one of these behaviours | 1 | |
| The rabbit does not present any of these behaviours | 2 | |
| *Pine cone, toy, pen substrate | | |
| **Food, vegetables, greens, or snacks. | | |
| **4) Facial Expression** | | |
| A) Keeps eyes wide open and ears erect all the time | S17 Video | https://youtu.be/qriFMIBaD1s |
| | S18 Video | https://youtu.be/6FXwm4wT13s |
| B) Keeps eyes semi-closed or closed at any time point* | S19 Video | https://youtu.be/JZtzJWY516E |
| | S20 Video | https://youtu.be/PXrX7sk4rT4 |
| C) Exhibits drooping ears at any time point | S21 Video | https://youtu.be/6KXCfNkRFj0 |
| | S22 Video | https://youtu.be/zox6Qi5VKiI |
| The rabbit displays expression A | 0 | |
| The rabbit displays expression B or C | 1 | |
| The rabbit displays expressions B and C | 2 | |
| *Blinking of eyes is not considered as semi-closed or closed eyes. | | |
| **5) Attention to the affected area** | | |
| A) Licks the affected area | S23 Video | https://youtu.be/H_9t9XXqomU |
| B) Presses the abdomen against the floor | S24 Video | https://youtu.be/v0kh4fkW0qA |
| C) Keeps one limb suspended | S25 Video | https://youtu.be/BY5VwWmDsLQ |

*(Continued)*

**Table 4.** (Continued)

| RABBIT PAIN BEHAVIOUR SCALE (RPBS) | | |
|---|---|---|
| The rabbit does not present any of these behaviours | 0 | |
| The rabbit presents one of these behaviours | 1 | |
| The rabbit presents more than one of these behaviours | 2 | |
| **6) Miscellaneous behaviours** | | |
| A) Attempts to stand up, but remains lying down | S26 Video | https://youtu.be/iEWMBLv97D0 |
| B) Rapid dorsal movement of the body (flinches) | S27 Video | https://youtu.be/gVXPTb4-H_w |
| C) Retracts and closes the eyes (winces) | S28 Video | https://youtu.be/ZgoAjA-oe6g |
| D) Tremors* | S29 Video | https://youtu.be/nFuAwRfGyw4 |
| The rabbit does not present any of these behaviours | 0 | |
| The rabbit presents one of these behaviours | 1 | |
| The rabbit presents more than one of these behaviours | 2 | |

*More easily observed in the head and ears

affected area" presented a significant loading value in this dimension, as this item had a loading value of 0.93 in Dimension 2. According to the confirmatory factor analysis comparing one dimension, two uncorrelated dimensions and two correlated dimensions, the parameters chi-square, comparative fit index, Tucker Lewis index, loglikelihood, root mean square error of approximation, Akaike and Bayesian information criterion were very similar among the three tested models, therefore, RPBS may be considered mathematically unidimensional [37].

## Distribution of scores

Score 0 was predominant at *baseline* for all items on the scale (Fig 2). Scores 1 or 2 were predominant at the time point of greatest pain intensity (*pain*) for all items. At *24 h post*, the distribution of scores was similar to the *baseline* time point, with the exception of attention to the affected area, where scores 1 and 2 were more frequent than at *baseline*.

## Intra-observer reliability

The intra-observer reliability was very good (>0.80) for all RPBS items, except for miscellaneous behaviours for observers 1 (moderate) and 2 (good) and facial expression for observer 2

**Table 5.** Loading values, eigenvalues, and variance of the RPBS items based on principal component analysis.

| Dimensions | 1 | 2 |
|---|---|---|
| **Items** | **Loading values** | |
| Posture | **0.89** | 0.03 |
| Activity | **0.85** | -0.24 |
| Interaction and appetite | **0.90** | -0.14 |
| Facial expression | **0.69** | 0.25 |
| Attention to the affected area | -0.03 | **0.93** |
| Miscellaneous behaviours | **0.63** | 0.27 |
| **Eigenvalue** | **3.18** | **1.08** |
| **Variance** | **53.06** | 18.06 |

The structure was determined considering items with a loading value $\geq$ 0.50 or $\leq$ -0.50 with representative dimension (eigenvalue > 1 and variance > 20%). The loading values in bold indicate the variables that contribute to each dimension and the respective accepted eigenvalue and variance [38].

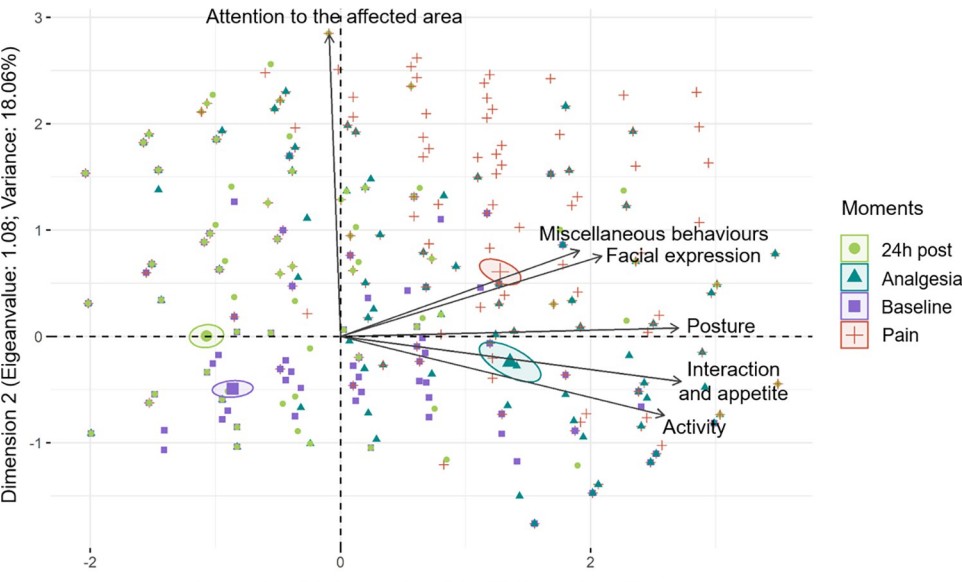

**Fig 1. Biplot for the principal component analysis with time points and items of the RPBS.** Ellipses were built according to perioperative time points of pain assessment. Time points: *baseline*—lilac, *pain*—red; *analgesia*–sea-green; *24h post*—green. The ellipses referring to the time when rabbits were in severe pain (*pain*) and after rescue analgesic (*analgesia*) were positioned at the right of the graph, where the vectors of items posture, activity, interaction and appetite, facial expression, and miscellaneous behaviours are also directed. The item attention to the affected area is in a different direction to the other items. The ellipses corresponding to the *baseline* and *24h post* time points are positioned in the opposite quadrant (left).

(good). Reliability was very good for the total sum of the RPBS ($\geq 0.91$) and NS and good or very good for rescue analgesic, SDS, and VAS (Table 6).

## Inter-observer reliability and matrix agreement of the RPBS

The inter-observer reliability of the unidimensional scales ranged from poor to very good. For RPBS, the reliability was very good among all evaluators ($\geq 0.86$) (Table 7).

Reliability between raters was good or very good for all RPBS items, except for the miscellaneous behaviour item, which ranged from fair to good (S4 Table).

## Criterion validity

**Concurrent criterion validity.** The positive correlations between the RPBS and NS, SDS, VAS, and RbtGS [18] were 0.86 ($p < 2.2^{-16}$), 0.80 ($p < 2.2^{-16}$), 0.84 ($p < 2.2^{-16}$), and 0.68 ($p < 2.2^{-16}$), respectively, therefore there was a high correlation between the RPBS and unidimensional scales and a moderate correlation between the RPBS and the RbtGS [46].

## Responsiveness

The total RPBS score was significantly higher at the *pain* time point compared to the other time points in all groups, which demonstrates the responsiveness of the instrument (Fig 3).

The evaluators (as a fixed effect) influenced the RPBS scores only in the evaluations of the Ortho group (p = 0.018).

Most of the RPBS items were responsive to pain in all groups (*baseline < pain*) and all of them were responsive for the Ortho group. All scales demonstrated responsiveness given that the highest scores were observed at the *pain* time point compared to *baseline*, however only

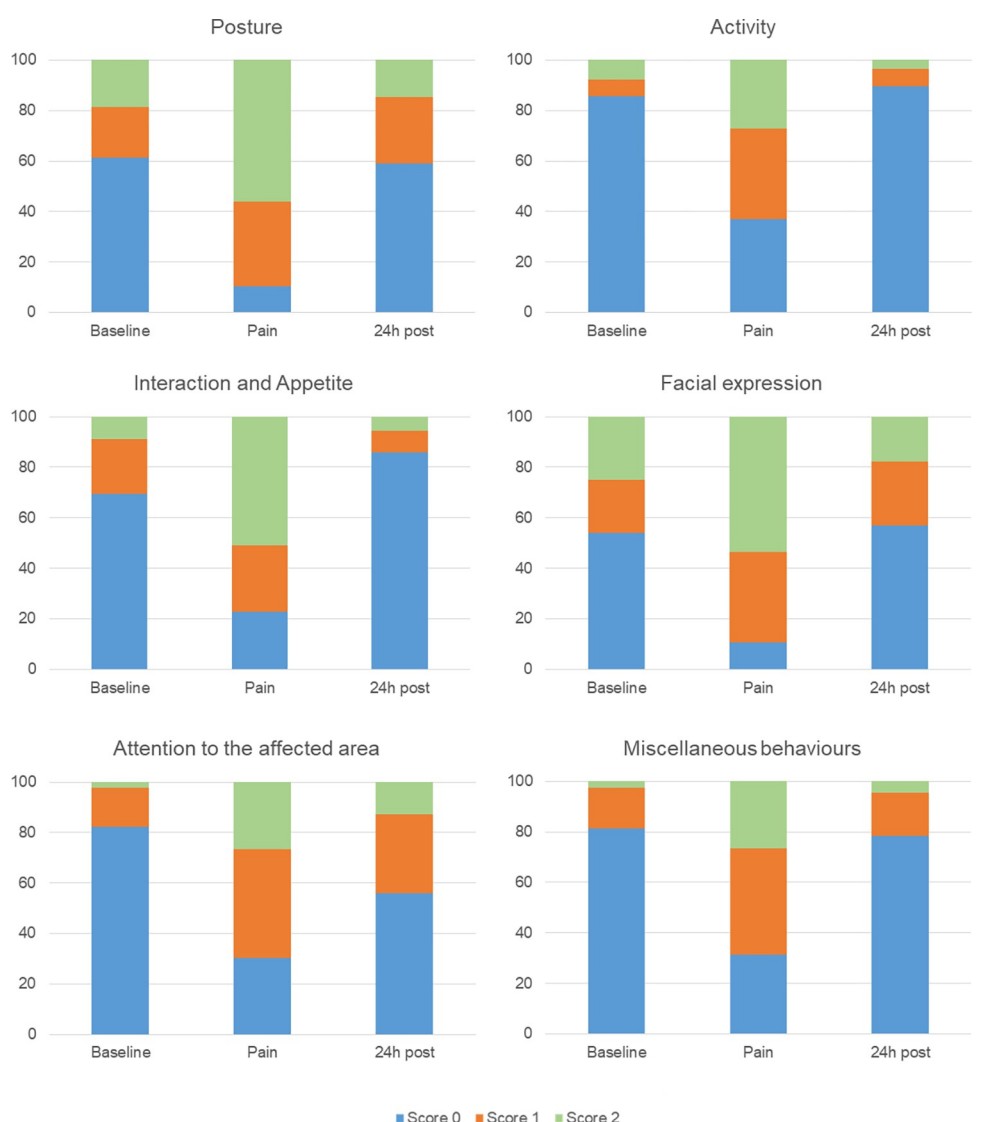

**Fig 2. Frequency of the presence of scores of each item of the RPBS.** Score 0 –blue, Score 1 –orange, Score 2 –green.

the RPBS detected differences between all time points in the Ortho group, that is, it was responsive to rescue analgesic and moderate pain at *24h post* (Table 8).

Considering each evaluated time point (*baseline*, *pain*, or *24h post*), the evaluators (fixed effect) influenced the RPBS scores at the three time points.

## Construct validity

The OVH-Pla and OVH-Melox groups presented significantly higher scores compared to the other groups at *baseline*. At the *pain* time point, the ORC-Multi group had significantly lower scores than the OVH-Pla, OVH-Melox, and Ortho groups. At *24h post*, the OVH-Pla group presented significantly higher scores than the other groups (Fig 4). Construct validity was confirmed by the three-hypothesis test, like reported for responsiveness, the RPBS score after surgery (*pain*) i. was higher than the preoperative score, ii. decreased after analgesia, and iii. over time.

**Table 6. Intra-observer reliability of RPBS, unidimensional scales, and rescue analgesia indication in rabbits.**

| Evaluator | 1 | 2 | 3 | 4 |
|---|---|---|---|---|
| **RPBS Items** | $k_w$ (min-max) | | | |
| Posture | **0.89 (0.85–0.93)** | **0.88 (0.84–0.92)** | **0.85 (0.80–0.89)** | **0.94 (0.91–0.96)** |
| Activity | **0.91 (0.91–0.91)** | **0.81 (0.77–0.85)** | **0.83 (0.83–0.83)** | **0.96 (0.96–0.96)** |
| Interaction and appetite | **0.95 (0.93–0.96)** | **0.96 (0.96–0.96)** | **0.89 (0.89–0.89)** | **0.97 (0.97–0.97)** |
| Facial expression | **0.85 (0.80–0.91)** | **0.71 (0.63–0.79)** | **0.84 (0.78–0.90)** | **0.93 (0.89–0.96)** |
| Attention to the affected area | **0.87 (0.85–0.90)** | **0.83 (0.83–0.83)** | **0.80 (0.80–0.80)** | **0.95 (0.95–0.95)** |
| Miscellaneous behaviours | 0.57 (0.56–0.58) | 0.61 (0.61–0.61) | **0.81 (0.78–0.85)** | **0.80 (0.80–0.80)** |
| RA | **0.80 (0.72–0.89)** | **0.73 (0.63–0.84)** | **0.81 (0.72–0.89)** | **0.87 (0.80–0.94)** |
| **Scales** | $k_w$ (min-max) | | | |
| Numerical scale (NS) | **0.94 (0.94–0.94)** | **0.86 (0.86–0.86)** | **0.84 (0.84–0.84)** | **0.83 (0.83–0.83)** |
| Simple descriptive scale (SDS) | **0.90 (0.88–0.93)** | **0.87 (0.85–0.90)** | **0.65 (0.65–0.65)** | **0.79 (0.79–0.79)** |
| | ICC (CI) | | | |
| RPBS | **0.93 (0.91–0.95)** | **0.94 (0.93–0.96)** | **0.95 (0.93–0.96)** | **0.97 (0.97–0.98)** |
| Visual analogue scale (VAS) | **0.94 (0.92–0.95)** | **0.87 (0.83–0.90)** | **0.82 (0.77–0.86)** | **0.81 (0.75–0.85)** |

RA—rescue analgesia indication. Each item of the RPBS, NS, and SDS was calculated using the kappa coefficient ($kw$); the sum of the RPBS and the VAS was calculated using intraclass correlation coefficient (ICC consistency), CI—Confidence interval. Interpretation of the degree of reliability $k_w$ or ICC (consistency): very good: 0.81–1.0; good: 0.61–0.80; moderate: 0.41–0.60; reasonable: 0.21–0.4; poor < 0.2 [15, 45, 48]. Bold type corresponds to values > 0.61.

Control rabbits filmed in the morning (n = 8, p < 0.0001), in the afternoon (n = 8, p < 0.0001), or in both periods grouped together (n = 16, p <0.0001) had lower RPBS scores compared to those filmed at similar *pain* time points, therefore confirming construct validity by using the known-group validity approach [27, 46].

Construct validity was further confirmed by the internal relationships among items according to principal component analysis, internal consistency and item-total correlation and by the high correlation between the RPBS and the unidimensional scales and moderate correlation with RbtGS.

## RPBS assessment throughout the day

There was no significant difference over time for the RPBS items and for the unidimensional scales in pain-free rabbits. The sum of the RPBS was higher at 2pm compared to 8am. The evaluators and the evaluation phase (as fixed effects) influenced the sum of the RPBS (Table 9).

**Table 7. Inter-observer reliability of RPBS, unidimensional scales, and rescue analgesia indication between observers.**

| Evaluators/ scales | kw (min-max) | | | ICC (CI) | |
|---|---|---|---|---|---|
| | Numerical scale (NS) | Simple descriptive scale (SDS) | Rescue analgesia (RA) | Rabbit Pain Behaviour Scale (RPBS) | Visual analogue scale (VAS) |
| Evaluator 1 vs 2 | 0.87 (0.87–0.87) | 0.86 (0.83–0.88) | 0.69 (0.62–0.76) | 0.90 (0.88–0.92) | 0.87 (0.85–0.90) |
| Evaluator 1 vs 3 | 0.67 (0.67–0.67) | 0.55 (0.49–0.61) | 0.74 (0.67–0.81) | 0.89 (0.86–0.90) | 0.64 (0.13–0.82) |
| Evaluator 1 vs 4 | 0.78 (0.78–0.78) | 0.65 (0.59–0.71) | 0.70 (0.63–0.77) | 0.90 (0.88–0.92) | 0.76 (0.62–0.84) |
| Evaluator 2 vs 3 | 0.66 (0.66–0.66) | 0.54 (0.48–0.60) | 0.77 (0.71–0.84) | 0.92 (0.90–0.93) | 0.63 (0.23–0.80) |
| Evaluator 2 vs 4 | 0.74 (0.74–0.74) | 0.64 (0.57–0.70) | 0.59 (0.52–0.67) | 0.93 (0.91–0.94) | 0.72 (0.61–0.79) |
| Evaluator 3 vs 4 | 0.67 (0.65–0.68) | 0.65 (0.58–0.71) | 0.71 (0.64–0.78) | 0.92 (0.91–0.94) | 0.67 (0.54–0.76) |

Each item of the RPBS, NS, and SDS was calculated with the kappa coefficient ($kw$); the sum of the RPBS and the VAS was calculated using intraclass correlation coefficient (ICC consistency), CI—Confidence interval. Interpretation of the degree of reliability $k_w$ or ICC (consistency): very good: 0.81–1.0; good: 0.61 12–0.80; moderate: 0.41–0.60; reasonable: 0.21–0.4; poor < 0.2 [45].

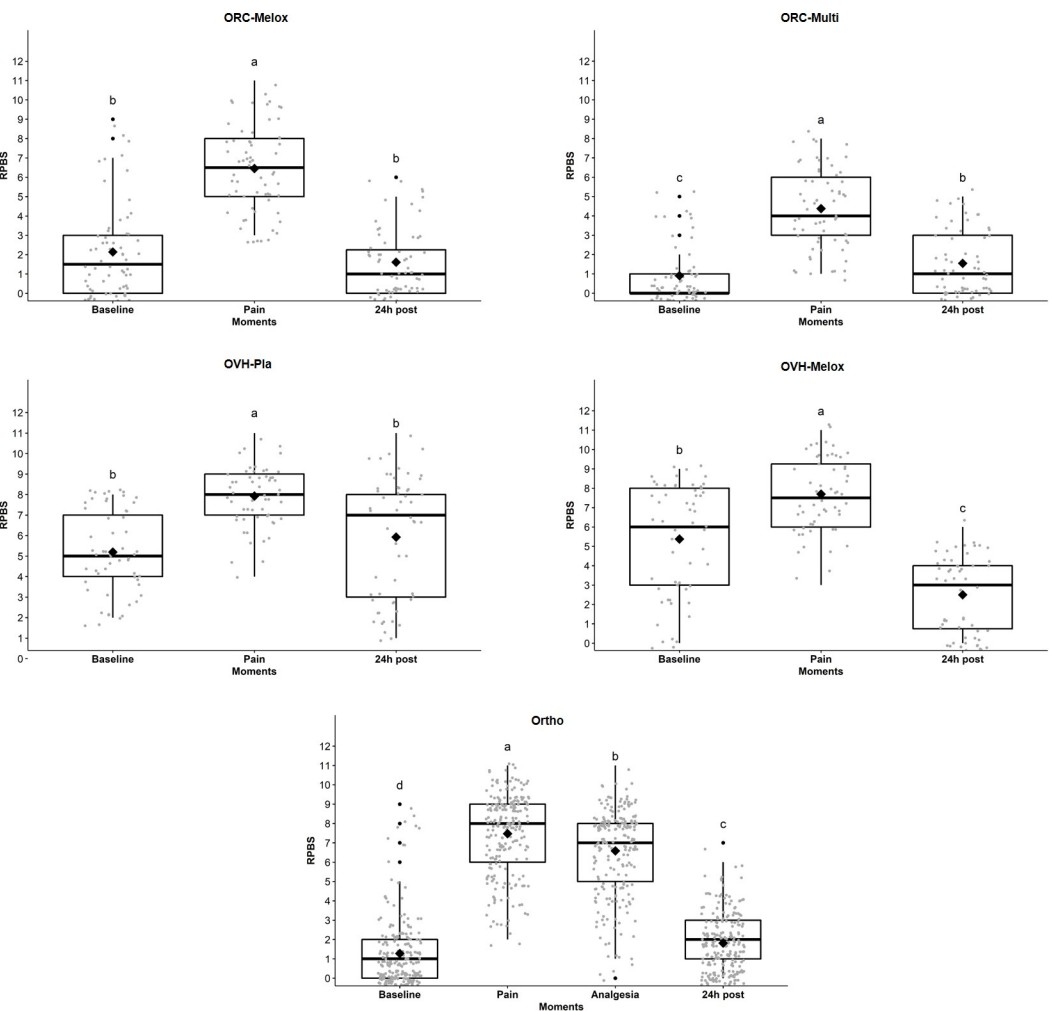

**Fig 3. Box-plots of the scores (median/amplitude) of the RPBS (rabbit pain behaviour scale), comparing the perioperative time points for each group.** The top and bottom box lines represent the interquartile range (25 to 75%), the line within the box represents the median, the extremes of the whiskers represent the minimum and maximum values, black lozenges (♦) represent the mean and black circles (•) represent outliers. RPBS: Rabbit pain behaviour scale. Groups: Ortho— 28 rabbits submitted to radio ostectomy; OVH-Pla—7 rabbits submitted to OVH under placebo administration; OVH-Melox– 7 rabbits submitted to OVH under meloxicam administration; ORC-Melox—8 rabbits submitted to orchiectomy under meloxicam administration; ORC-Multi– 8 rabbits submitted to orchiectomy under the administration of a multimodal analgesic protocol. Different letters express significant differences between time points where a > b > c > d, according to the mixed linear model.

None of the rabbits at 8am, 11% of the rabbits at 2pm, and 9% of the rabbits at 8pm would receive rescue analgesic according to the Youden index (YI).

## Item-total correlation, internal consistency, specificity and sensitivity

All items presented an adequate item-total Spearman correlation coefficient ($> 0.3$), except attention to the affected area. The internal consistency of the scale was very good (Table 10).

With the exception of posture and facial expression, the other RPBS items were specific, i.e infrequently observed at *baseline*. With the exception of activity, the RPBS items were sensitive, i.e frequent observed when rabbits were suffering pain (Table 10).

**Table 8. Pain scores of the RPBS, rescue analgesia, and unidimensional pain scales between the time points evaluated for each group.**

| Items RPBS | Groups | Time points | | | |
|---|---|---|---|---|---|
| | | Baseline | Pain | Analgesia | 24h post |
| Posture | ORC-Melox | 0 (0–2)$^b$ | 2 (0–2)$^a$ | | 0 (0–2)$^b$ |
| | ORC-Multi | 0 (0–2)$^b$ | 1 (0–2)$^a$ | | 0 (0–2)$^{ab}$ |
| | OVH-Pla | 2 (0–2) | 2 (1–2) | | 2 (0–2) |
| | OVH-Melox | 2 (0–2)$^a$ | 2 (1–2)$^a$ | | 1 (0–2)$^b$ |
| | Ortho | 0 (0–2)$^b$ | 2 (0–2)$^a$ | 2 (0–2)$^a$ | 0 (0–2)$^b$ |
| Activity | ORC-Melox | 0 (0–2)$^b$ | 0 (0–2)$^a$ | | 0 (0–1)$^b$ |
| | ORC-Multi | 0 (0–1) | 0 (0–1) | | 0 (0–1) |
| | OVH-Pla | 0 (0–2)$^b$ | 1 (0–2)$^a$ | | 0 (0–2)$^{ab}$ |
| | OVH-Melox | 0 (0–2)$^a$ | 1 (0–2)$^a$ | | 0 (0–2)$^b$ |
| | Ortho | 0 (0–2)$^b$ | 1 (0–2)$^a$ | 2 (0–2)$^a$ | 0 (0–1)$^b$ |
| Interaction and appetite | ORC-Melox | 0 (0–2)$^{ab}$ | 0 (0–2)$^a$ | | 0 (0–0)$^b$ |
| | ORC-Multi | 0 (0–1)$^{ab}$ | 0 (0–1)$^a$ | | 0 (0–1)$^b$ |
| | OVH-Pla | 1 (0–2)$^b$ | 2 (0–2)$^a$ | | 1 (0–2)$^{ab}$ |
| | OVH-Melox | 1 (0–2)$^a$ | 1 (0–2)$^a$ | | 0 (0–0)$^b$ |
| | Ortho | 0 (0–2)$^b$ | 2 (0–2)$^a$ | 2 (0–2)$^a$ | 0 (0–1)$^b$ |
| Facial expression | ORC-Melox | 1 (0–2)$^b$ | 2 (0–2)$^a$ | | 0 (0–2)$^b$ |
| | ORC-Multi | 0 (0–2)$^b$ | 2 (0–2)$^a$ | | 0 (0–2)$^b$ |
| | OVH-Pla | 2 (0–2)$^a$ | 2 (1–2)$^a$ | | 2 (0–2)$^b$ |
| | OVH-Melox | 2 (0–2)$^a$ | 2 (1–2)$^a$ | | 1 (0–2)$^b$ |
| | Ortho | 0 (0–2)$^b$ | 1 (0–2)$^a$ | 1 (0–2)$^a$ | 0 (0–2)$^b$ |
| Attention to the affected area | ORC-Melox | 0 (0–1)$^c$ | 1 (0–2)$^a$ | | 0 (0–1)$^b$ |
| | ORC-Multi | 0 (0–1)$^b$ | 1 (0–2)$^a$ | | 0 (0–1)$^b$ |
| | OVH-Pla | 0 (0–2)$^b$ | 1 (0–2)$^a$ | | 1 (0–2)$^{ab}$ |
| | OVH-Melox | 0 (0–1)$^b$ | 1 (0–2)$^a$ | | 0 (0–2)$^b$ |
| | Ortho | 0 (0–2)$^c$ | 1 (0–2)$^a$ | 0 (0–2)$^b$ | 1 (0–2)$^b$ |
| Miscellaneous behaviours | ORC-Melox | 0 (0–2)$^b$ | 1 (0–2)$^a$ | | 0 (0–2)$^b$ |
| | ORC-Multi | 0 (0–2)$^b$ | 1 (0–2)$^a$ | | 0 (0–2)$^{ab}$ |
| | OVH-Pla | 0 (0–2)$^b$ | 1 (0–2)$^a$ | | 0 (0–2)$^{ab}$ |
| | OVH-Melox | 1 (0–2)$^b$ | 1 (0–2)$^a$ | | 0 (0–1)$^b$ |
| | Ortho | 0 (0–1)$^c$ | 1 (0–2)$^a$ | 0 (0–2)$^b$ | 0 (0–2)$^c$ |
| **RPBS** | ORC-Melox | 2 (0–9)$^b$ | 7 (3–11)$^a$ | | 1 (0–6)$^b$ |
| | ORC-Multi | 0 (0–5)$^c$ | 4 (1–8)$^a$ | | 1 (0–5)$^b$ |
| | OVH-Pla | 5 (2–8)$^b$ | 8 (4–11)$^a$ | | 7 (1–11)$^b$ |
| | OVH-Melox | 6 (0–9)$^b$ | 8 (3–11)$^a$ | | 3 (0–6)$^c$ |
| | Ortho | 1 (0–9)$^d$ | 8 (2–11)$^a$ | 7 (0–11)$^b$ | 2 (0–7)$^c$ |
| **Rescue analgesia (RA)** | ORC-Melox | 0 (0–1)$^b$ | 1 (0–1)$^a$ | | 0 (0–1)$^b$ |
| | ORC-Multi | 0 (0–1)$^c$ | 1 (0–1)$^a$ | | 0 (0–1)$^b$ |
| | OVH-Pla | 1 (0–1)$^b$ | 1 (0–1)$^a$ | | 1 (0–1)$^{ab}$ |
| | OVH-Melox | 1 (0–1)$^a$ | 1 (1–1)$^{ab}$ | | 1 (0–1)$^b$ |
| | Ortho | 0 (0–1)$^c$ | 1 (0–1)$^a$ | 1 (0–1)$^a$ | 0 (0–1)$^b$ |
| **Numerical scale (NS)** | ORC-Melox | 1 (1–7)$^b$ | 6 (2–10)$^a$ | | 2 (1–7)$^b$ |
| | ORC-Multi | 1 (1–5)$^b$ | 5 (1–9)$^a$ | | 2 (1–7)$^b$ |
| | OVH-Pla | 6 (1–10)$^b$ | 8 (4–10)$^a$ | | 6 (1–10)$^{ab}$ |
| | OVH-Melox | 6 (1–10)$^b$ | 7 (3–10)$^a$ | | 4 (1–7)$^c$ |
| | Ortho | 1 (1–10)$^b$ | 7 (2–10)$^a$ | 7 (1–10)$^a$ | 3 (1–7)$^c$ |

(*Continued*)

**Table 8.** (Continued)

| Items RPBS | Groups | Time points | | | |
|---|---|---|---|---|---|
| | | Baseline | Pain | Analgesia | 24h post |
| **Simple descriptive scale (SDS)** | ORC-Melox | 1 (1–3)[b] | 3 (1–4)[a] | | 2 (1–3)[b] |
| | ORC-Multi | 1 (1–3)[c] | 2 (1–4)[a] | | 1 (1–3)[b] |
| | OVH-Pla | 3 (1–4)[b] | 3 (2–4)[a] | | 3 (1–4)[ab] |
| | OVH-Melox | 3 (1–4)[b] | 3 (2–4)[a] | | 2 (1–4)[c] |
| | Ortho | 1 (1–4)[b] | 3 (2–4)[a] | 3 (1–4)[a] | 2 (1–3)[c] |
| **Visual Analogue Scale (VAS)** | ORC-Melox | 0 (0–71)[b] | 60 (6–100)[a] | | 8 (0–67)[b] |
| | ORC-Multi | 0 (0–51)[c] | 40 (0–94)[a] | | 4 (0–61)[b] |
| | OVH-Pla | 56 (0–100)[b] | 73 (32–100)[a] | | 57 (0–100)[ab] |
| | OVH-Melox | 60 (0–100)[b] | 70 (30–100)[a] | | 20 (0–80)[c] |
| | Ortho | 0 (0–100)[b] | 71 (9–100)[a] | 65 (0–100)[a] | 10 (0–70)[c] |

RPBS–Rabbit pain behaviour scale; RA (0—no; 1—yes); NS (1–10), SDS (1–4), and VAS (0–100). Different letters express significant differences between time points where a > b > c > d, according to the mixed linear model [24] for the RPBS and Friedman and Dunn post-test for the others.

### ROC curve, Youden index, cut-off point, and diagnostic uncertainty zone

The Youden index determined a score $\geq 3$ as the cut-off point to distinguish rabbits in pain from those without pain. The intervals between the sensitivity and specificity values of 0.90 were from 2.5 to 2.9. The 95% confidence intervals replicating the original ROC curve 1,001 times by the bootstrap method were between 2.5 and 3.5. Based on the last method, which presented the largest interval, the diagnostic uncertainty zone is between 3 and 4. The area under the curve value of 0.954 demonstrates an excellent discriminatory capacity of the RPBS (Fig 5).

The cut-off point determined by the Youden Index was $\geq 4$ of 10 for NS, $\geq 2$ of 4 for SDS, and $\geq 23$ of 100 for VAS (Table 11).

The percentage of assessments scored in the diagnostic uncertainty zone (3 and 4) was low at *baseline* (14%), *pain* (13%), and *analgesia* time points (13%), and at *24h post* it was slightly greater than the other time points (24%) (Table 12).

**Predictive criterion validity.** According to the Youden index and the need for rescue analgesia according to the experience of the evaluator, in both cases 97% of the rabbits would receive analgesia at the time point considered to be associated with greatest pain after surgery (sensitivity) and 34% and 29% of the rabbits would receive unnecessary analgesia before the surgical procedure (specificity) respectively, which indicates an excellent predictive criterion validity when rabbits experience pain. However, when considering each group separately, 88% and 79% of rabbits submitted to OVH would receive unnecessary analgesia at *baseline* according to the Youden Index and to the evaluator's experience respectively (Table 13). A minority of rabbits would receive rescue analgesia at *baseline* for the other groups. At *pain* time points more than 80% of rabbits would receive analgesia in all groups.

## Discussion

This study is pioneering in terms of the robust methods used to validate a pain scale in rabbits undergoing orthopaedic and soft tissue surgeries, based on COSMIN guidelines [47]. This is confirmed by content, criterion, and construct validity and reliability in recognizing postoperative pain in rabbits, and the determination of an intervention point (cut-off point that identifies pain and support treatment). The variety of surgical stimuli used included different qualities and intensities of pain.

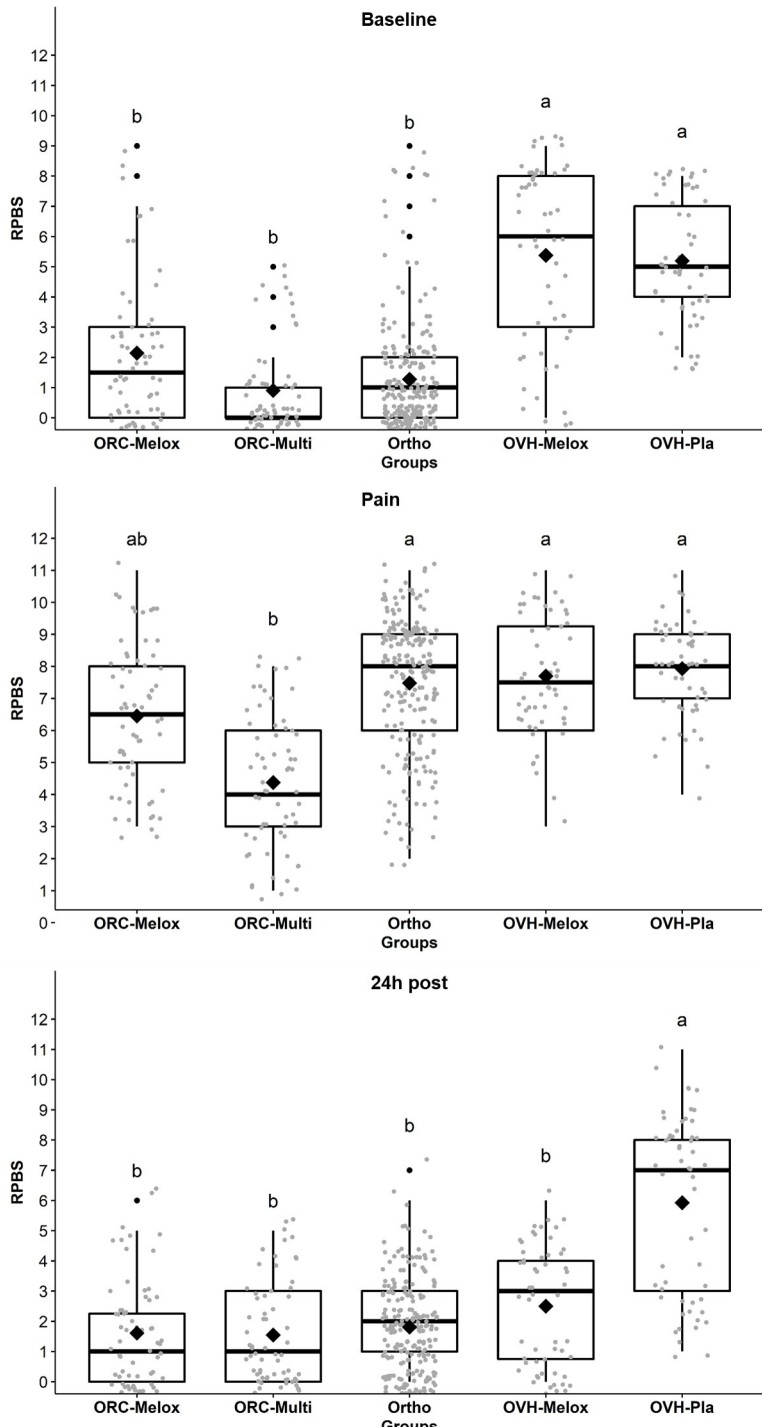

**Fig 4. Box-plot of the scores (median/amplitude) of the RPBS (rabit pain behaviour scale), comparing the groups at each perioperative time point.** The top and bottom box lines represent the interquartile range (25 to 75%), the line within the box represents the median, the extremes of the whiskers represent the minimum and maximum values, black lozenges (♦) represent the mean and black circles (•) represent outliers. RPBS: Rabbit pain behaviour scale. Different letters express significant differences between groups where a > b, according to the mixed linear model [24].

**Table 9. Evaluation of RPBS and unidimensional scales between three different periods of the day.**

| Item/Scale | Time point | | |
|---|---|---|---|
| | **8am** | **2pm** | **8pm** |
| **Posture** | 0 (0–1) | 0 (0–2) | 0 (0–1) |
| **Activity** | 0 (0–0) | 0 (0–1) | 0 (0–1) |
| **Interaction and appetite** | 0 (0–0) | 0 (0–1) | 0 (0–1) |
| **Facial expression** | 0 (0–1) | 0 (0–2) | 0 (0–1) |
| **Attention to the affected area** | 0 (0–1) | 0 (0–1) | 0 (0–2) |
| **Miscellaneous behaviours** | 0 (0–0) | 0 (0–1) | 0 (0–0) |
| **RPBS** | 0 (0–2)[b] | 0 (0–6)[a] | 0 (0–5)[ab] |
| **NS** | 1 (1–3) | 1 (1–7) | 1 (1–3) |
| **SDS** | 1 (1–2) | 1 (1–3) | 1 (1–3) |
| **VAS** | 0 (0–50) | 0 (0–69) | 0 (0–60) |

RPBS–Rabbit pain scale; NS (1–10), SDS (1–4), and VAS (0–100). Different letters express significant differences between time points where a > b, according to the mixed linear model for RPBS and Friedman and Dunn post-test for the others.

The original selection of behaviours in the scale was based on the ethogram [12], literature [11, 12, 18, 29–34], and judgment of an expert committee. This supports the instrument's content validity, which confirmed that the items reflect the phenomenon being evaluated, i.e., pain [50, 56, 58, 59]. Subsequently, the sub-item "spasms", although relevant in the literature [11] and in the content validity, did not meet the refinement criteria and was, therefore, the only sub-item excluded. This reflects the need to submit new instruments to strict statistical refinement criteria to increase its discriminatory capacity and avoid the insertion of unnecessary items. The RPBS items are similar to those approved in five phases during the creation of the BRPS [20], a scale developed in parallel and published after completion of the current study. The RPBS is based on strict and complete behaviour selection criteria, which confirms its importance in identifying pain in the species. The items locomotion, posture, ears/eyes and grooming in the BRPS are similar to activity, posture, facial expression, interaction and appetite in the RPBS. Both studies used OVH and orchiectomy as pain models, like in other species [15, 22, 23, 60], however only the RPBS included an orthopaedic surgery group. The different items between the two scales were the presence of the demeanor item in the BRPS and the items attention to the affected area and miscellaneous behaviours in the RPBS. The BRPS total score ranges from 0 to 21 and was validated in pet rabbits, otherwise the RPBS total score ranges from 0 to 12, was validated in research rabbits and included intervention score, i.e. a score indicating the need for rescue analgesia.

In addition to refinement, the homogeneity of the evaluators' training and experience reduces data variability and increases the validation and reliability of a new instrument [61, 62]. It is possible that the training led to high intra and inter-observer reliability [45]. The CANCRs scale [19] and the RbtGS [18] also presented very good inter-observer reliability, however, the authors did not assess intra-observer reliability. This is an important attribute to determine the repeatability of measurements, as this represents the consistency of scores over time [21, 58, 63]. The recently published BRPS [20] also did not assess repeatability and reproducibility.

Principal component analysis is a multivariate analysis used to explore the dimensionality and the multiple interactions among the items of a scale, by segregating the correlated items in the same dimension or principal component. The magnitude of the correlation of each item

**Table 10. Item-total correlation, internal consistency, specificity, and sensitivity of the RPBS.**

| Item | Item-total correlation ($r_s$) | Internal consistency | | Specificity (Sp) % | Sensitivity (S) % |
|---|---|---|---|---|---|
| | | Cronbach's α | Mcdonald's ω | | |
| | | 0.78 | 0.83 | | |
| | RPBS | Excluding each item below | | RPBS | |
| Posture | **0.77** | **0.71** | **0.76** | 61 | **90** |
| Activity | **0.69** | **0.68** | **0.78** | 86 | 63 |
| Interaction and appetite | **0.74** | **0.74** | **0.76** | 70 | 77 |
| Facial expression | **0.57** | **0.85** | **0.82** | 54 | **89** |
| Attention to the affected area | 0.04 | **0.75** | **0.88** | 82 | 70 |
| Miscellaneous behaviours | **0.49** | **0.78** | **0.83** | 81 | 69 |

RPBS: Rabbit pain behaviour scale. Degree of correlation $r_s$: the items were accepted when Spearman's rank correlation coefficient was > 0.3 (bold) [56]. Cronbach's α coefficient and McDonald's ω coefficient were calculated for the total score and excluding each item from the scale. Interpretation of the α coefficient values: 0.60–0.64 minimally acceptable; 0.65–0.69 acceptable; 0.70–0.74 good; 0.75–0.80 very good; and > 0.80 excellent [50, 56, 57]. McDonald's omega coefficient interpretation: 0.65–0.80, acceptable; >0.80 strong reliability evidence [49]. Acceptable values are highlighted in bold (> 0.65). Interpretation of specificity (Sp) and sensitivity (S): excellent 95–100%; good 85–94.9%; moderate 70–84.9%; not specific or sensitive < 70%; bold values ≥ 70% [48].

with a given dimension is determined by the loading value [64]. The items posture, activity and interaction, and appetite showed the highest loading values, therefore the greatest variability and importance. The items facial expression and miscellaneous behaviours showed lower loading values, but remained within limits determined in our study (> 0.50 or < -0.50), and therefore contributing to the first dimension. The loading value of attention to the affected area was below the limit for inclusion in the first dimension. Although this item had a higher loading value in the second dimension, this dimension was not retained in our study according to the methods used.

There are different ways to investigate dimensionality and multiple association between variables according to exploratory and/or confirmatory factor analysis. Principal component analysis, an exploratory analysis, was used herein to maintain consistency with previously

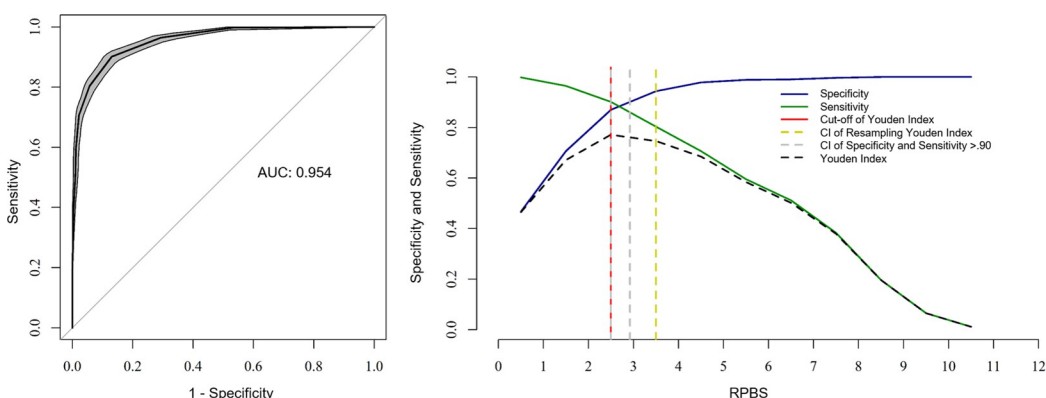

**Fig 5. ROC curve and AUC [left] and ROC curve of two graphs with the diagnostic uncertainty zone for the RPBS [right].** ROC curve (receiver operating characteristic) with a 95% confidence interval (CI) calculated from 1,001 replications and area under the curve (AUC) [left]. Interpretation of AUC ≥ 0.95—high discriminatory capacity. ROC curve of two graphs, CI of 1,001 replications and sensitivity and specificity > 0.90 applied to estimate the diagnostic uncertainty zone of the cut-off point, according to the Youden index for the Rabbit pain behaviour scale (RPBS) [right] [54, 55]. The diagnostic uncertainty zone scores ranged from 2.5 (3) to 3.5 (4); therefore, < 3 indicates truly negative pain (rabbit without pain) and ≥ 4 indicates truly positive pain (rabbit suffering pain). A Youden index ≥ 3 represents the cut-off point for indication of rescue analgesia [15].

**Table 11. Scores, specificity, sensitivity, and Youden index corresponding toindication for rescue analgesia of the RPBS and unidimensional scales.**

| Scale | Score | Specificity | Sensitivity | Youden index |
|---|---|---|---|---|
| RPBS | 3 | 0.87 | 0.90 | 0.77 |
| NS | 4 | 0.96 | 0.87 | 0.83 |
| SDS | 2 | 0.77 | 1 | 0.76 |
| VAS | 23 | 0.97 | 0.88 | 0.85 |

Scales: RPBS–Rabbit pain behaviour scale; NS–numerical; SDS—simple descriptive; VAS—visual analogue.

validated scales in other species [15, 22–25]. According to PCA all items pertained to the first dimension, except attention to the affected area, which pertained to the second dimension. Due to the PCA indicating that the RPBS is potentially bidimensional, data were further processed for confirmatory factor analysis (data not shown), Horn's Parallel Analysis, Marchenko-Pastur limit and Gavish-Donoho method to compare a one-dimensional model against a two-dimensional model. To proceed confirmatory factor analysis, it is recommended to include, in the structural models, more than three variables in each dimension to avoid calculation error. Except for the Horn's Parallel Analysis which suggested two dimensions, Marchenko-Pastur limit, Gavish-Donoho method and confirmatory factor analysis showed no improvement by using a two-dimensional model, which supports the conclusion that the one-dimension model offers the best fit for RPBS.

RPBS is mathematically unidimensional, but biologically multidimensional, as it assesses other dimensions in addition to pain intensity [65], such as temporal (score changes after surgery and after rescue analgesia) and qualitative characteristics of pain: sensory (attention to the affected area and miscellaneous behaviour), affective (interaction with motivational items, exploration), cognitive (posture and activity) and physiological (appetite) dimensions.

The different scores (0, 1, or 2) assigned to each item on the scale were distributed as expected. Scores "0" predominated at *baseline* for all scale items and scores 1 and 2 predominated after surgery, that is, at the time of greatest expected pain. At *24h post*, the scores were similar to the *baseline* time point, except for the item "attention to the affected area", which presented increased scores after surgery, but did not return to *baseline* values after 24 hours, possibly due to the residual nociceptive stimulus produced by local inflammation.

Concurrent and predictive criterion validity are crucial attributes in the validation of a new instrument [23–25]. The concurrent criterion validity assesses whether the new instrument is comparable to an established method with the same aim [63, 66]. For this purpose, the RPBS was compared to the unidimensional scales VAS, SDS, and NS, as described for other species [15, 22–24], and to the facial scale (RbtGS) [18], the only instrument available to assess pain in

**Table 12. Percentage of rabbits present in the diagnostic uncertainty zone according to the Youden index of the RPBS.**

| Evaluator \ Time point | Baseline | Pain | Analgesia | 24h post | MG |
|---|---|---|---|---|---|
| 1 | 17 | 21 | 23 | 33 | **24** |
| 2 | 17 | 9 | 7 | 28 | **17** |
| 3 | 10 | 10 | 14 | 13 | **12** |
| 4 | 11 | 9 | 9 | 21 | **13** |
| All | 14 | 13 | 13 | 24 | **16** |

Calculation based on 58 rabbits evaluated twice by four evaluators. RPBS: Rabbit pain behaviour scale MG—data of grouped time points (*baseline+pain+analgesia+24h post*). The diagnostic uncertainty zone was 3 to 4; < 3 indicates pain-free rabbit (true negative) and ≥ 4 indicates rabbit suffering pain (true positive).

**Table 13. Percentage of rabbits in each group would receive rescue analgesia in *baseline* and *pain* time points, according to the Evaluator Experience (EE) and the Youden Index (YI) of the RPBS.**

| Groups | RA according to the | Baseline | Pain |
|---|---|---|---|
| ORC-Melox (n = 8) | YI | 34.4 | 100 |
| | EE | 28.1 | 98.4 |
| ORC-Multi (n = 8) | YI | 15.6 | 79.7 |
| | EE | 6.3 | 85.9 |
| OVH-Pla (n = 7) | YI | 87.5 | 100 |
| | EE | 75.0 | 98.2 |
| OVH-Melox (n = 7) | YI | 78.6 | 100 |
| | EE | 78.6 | 100 |
| Ortho (n = 28) | YI | 14.3 | 98.7 |
| | EE | 11.6 | 98.2 |

the leporine species at the time this study was accomplished (2018–2019). In this sense, the RPBS showed a high correlation with the unidimensional scales. However, the correlation with the facial scale (RbtGS) [18] was only moderate, similar to the validation of the scale to assess acute pain in sheep [24]. This result may be related to the lack of evaluators' training to use the RbtGS and/or the lack of refinement of the latter instrument, which did not pass-through similar validation criteria (criterion validity, intra-rater reliability, principal component analysis and internal consistency).

Responsiveness was demonstrated by the change in the scale scores over time, with higher scores given after surgical stimuli compared to *baseline* for all groups. For the Ortho group, there was still a reduction in pain scores after analgesia and at 24 hours. Responsiveness refers to the ability of the instrument to detect a significant change in a clinical state [46]. We believe that the period of time and the analgesic regimen used to check the effectiveness of rescue analgesia was sufficient as a similar dose of morphine (3mg/kg), to that used in the current study (2 mg/kg), increased the thermal threshold in rabbits for 4 hours [67] and meloxicam was effective to treat pain after soft tissue surgery in rabbits [11]. A similar period and analgesic protocol composed of anti-inflammatories and opioids, have also been used to assess responsiveness in previously published animal pain scales [15, 22–25] at 2 [23, 25] and 4 hours [15] after surgery.

Construct validity was confirmed by four approaches according to COSMIN methodology: the three-hypothesis test, known-group validity, internal relationships among items and correlation with other pain assessment instruments [26, 47].

Time of day influences pain behavioural expression in some species such as horses [68] and rabbits [11]. In the afternoon, the median scores of RPBS were higher than in the morning for the ORTHO group. However, according to the Youden Index, the recommendation for rescue analgesic would be infrequent and similar for the three periods (≤11%) and most scores were within the diagnostic uncertainty zone (scores 3 and 4), which represents a homogeneous specificity of the scale throughout the day.

The high RPBS scores at *baseline* in the OVH-Pla and OVH-Melox groups, could indicate that these animals were mistakenly identified as experiencing pain [11]. A possible explanation for the difference in scores at *baseline* between the groups is the methodological design. Motivational items (pinecones and carrots) were offered at the beginning of each recording only in rabbits undergoing orthopaedic surgery. This environment enrichment stimulated activity, interaction, and movement in the rabbits without pain [12] and improved the specificity of the scale. This result suggests the potential value of including stimulating items such as palatable

food and a new toy before employing the RPBS. However, the presence of motivational items does not solely explain the high *baseline* RPBS scores in the OVH-Pla and OVH-Melox groups because the groups undergoing orchiectomy showed good specificity for the RPBS (low scores), even without the presence of stimulating items. Considering that both male (ORC) and female (OVH) were exposed to the same conditions, a speculative explanation is that the lack of environmental enrichment accentuated the greater activity of males than in females [69], but this requires further investigation.

The predictive criterion validity of the RPBS was further supported by the Youden Index, as 97% of rabbits would receive rescue analgesia at the time associated with the greatest pain after surgery thus guaranteeing pain relief. However, at *baseline* about one third of rabbits would have received unnecessary analgesia according to the Youden Index, with the vast majority of these belonging to groups submitted to OVH. The two possible explanations for this finding are possibly related to the lower activity of rabbits in the morning (*baseline*) and lack of environmental enrichment, therefore reducing activity in these groups, but again this requires further investigation.

The RPBS was potentially sensitive enough to differentiate between moderate and severe pain as after surgery male rabbits castrated with a multimodal analgesic protocol (ORC-Multi) had lower scores than those treated only with an anti-inflammatory (ORC-Melox), and those submitted to more invasive surgical procedures (OVH-Melox, OVH-Pla and Ortho) [70]. Additionally, 24 hours after surgery, pain scores were higher in the OVH-Pla group compared to other groups, probably due to hypersensitization due to the lack of analgesia, which demonstrates the sensitivity of the scale to different surgeries and analgesic treatments.

The item-total correlation estimates the homogeneity of the scale by independently correlating each item on the scale with the total score after excluding the item evaluated. If the correlation drops significantly ($r_s < 0.3$) when the item in question is excluded, it is because the item has little correlation with the scale as a whole [56]. Most of the RPBS items presented an adequate correlation ($r_s > 0.3$ and $< 0.7$), which suggests that they contribute to the total scale score [48]. Posture and interaction showed a correlation slightly higher than 0.7, which may characterize their redundancy [56] and attention to the affected area had a very low item-total correlation. These three items were maintained as they were approved by the other refinement criteria.

Internal consistency showed that the scale items behave in the same way, with a concomitant increase or decrease in their scores according to the intensity of pain [50]. Our values were similar but a little lower than the BRPS [20].

Most of the items were specific, which supports the correct identification of the true negatives, that is, the rabbits that do not present pain, except for posture and facial expression. Almost all items on the scale were sensitive, i.e., identifies true positives (rabbits in pain). Ideally, a scale should have high specificity and sensitivity to avoid unnecessary analgesic treatment in pain-free animals [71] and provision of analgesia to animals in pain respectively.

The scales previously developed to assess pain in rabbits [18–20] did not determine a cut-off point for rescue analgesia. In the present study, the RPBS cut-off point was set at $\geq 3$, with a high area under the curve representing the scale's excellent discriminatory ability to identify pain [72]. There was a gray zone or diagnostic uncertainty [73] corresponding to scores 3 and 4, an interval that includes false negative and false positive scores, so rabbits that present a score $\geq 4$ postoperatively require analgesic intervention with a greater degree of certainty. Although few evaluations corresponded to scores within this zone, the highest frequency occurred 24 hours after surgery. This is likely due to greater individual variation, types of surgery and analgesia, and moderate degree of pain.

This study is not without limitations. The original recordings had different durations for the three types of surgeries. To compensate for a possible confirmation bias [74] caused by shortening the original videos to 2–3 minute duration, these sequences were objectively based on proportional duration and frequency of the behaviours displayed in the original videos according to the ethogram. To minimize subjective expectation bias, the evaluators scored the videos in a random order, and they were unaware of the time points of the perioperative period. However, subjective and confirmatory bias cannot be completely ruled out since the evaluators were aware of the objective of the study. There was a possible information bias because the implementation of motivational items at the time of assessment was not standardized in all types of surgery, apparently favoring the specificity data in rabbits undergoing orthopaedic surgery. The positive side of this bias was the demonstration of the importance of using motivational items in assessments. Another limitation was that the RPBS was validated for healthy laboratory rabbits, which requires clinical validation of the scale to assess pain in pet rabbits of different breeds, ages, and under different pain conditions, including clinical and chronic pain such as described for CANCRs [19] A third limitation was that the evaluators went through training and had the same education and similar time of experience. Some studies have shown that pain assessment is influenced by previous experience [75, 76]. Assessing how observers with different backgrounds and cultures assess the scale was not the aim of this study. This is an initial study for the validation of the RPBS, which should be subsequently submitted to cross-cultural validity [26], including evaluators with different backgrounds and ethnicities or languages, as performed for other scales after their validation [77, 78]. The evaluators who use the RPBS should carry out prior training to ensure the reliability and validity of the results until a further evaluation is performed with the RPBS used by evaluators without training.

In this study, like in the study that developed the facial expression-based scale [18], the evaluators only observed the images and descriptors of the FAUs prior to starting the assessments. Although the correlation of the RPBS with the unidimensional scales was high even without evaluators' training, the lack of training before using RbtGS may have impaired its correlation with the RPBS. In rats, training has increased reliability when assessing pain by facial expression [61]. Still, the correlation between behaviours and a facial scale used for pain assessment was inadequate even when evaluated by trained raters [79].

In summary, the RPBS is a simple, effective, and practical tool to assess pain. It does not require interaction and handling of the rabbit, or taking parameters that may interfere with the behavioural responses [19]. It defines a score to institute rescue analgesic, which facilitates decision-making about the qualitative and quantitative treatment for postoperative pain, ensuring the well-being of these animals in veterinary clinics and research facilities. As this was a bicentric study, the scale was versatile for different types of accommodation (pens or cages), anaesthetic and analgesic protocols, surgery types and degree of surgical invasiveness. Given the successful validation of this instrument, the authors suggest that the RPBS, or another previously published instruments [18–20] be part of research protocols in rabbits that are likely to be associated with pain, to improve well-being and avoid rabbits in pain not receiving analgesia (i.e., oligo-analgesia) [2–4].

It is concluded that after refinement and according to the validation criteria used, RPBS presents adequate repeatability, reproducibility, item-total correlation, internal consistency, responsiveness, and content, criterion and construct validity, as well as an intervention point (cut-off point) for rescue analgesia in the assessment of postoperative pain in rabbits undergoing orthopaedic and soft tissue surgery. The use of motivational items is recommended to minimize false positive pain diagnosis since they stimulate activity, interaction, and movement of the pain-free rabbits, improving the specificity of the scale.

## Supporting information

**S1 Table. First version of the scale (pre-refinement).**
(DOCX)

**S2 Table. Criteria used to select the behaviours included in the pre-refined RPBS used for video analysis based on content validity and behaviours reported in the literature.** Items with a total score $> 0.5$ were included in the scale.
(DOCX)

**S3 Table. Refinement process for inclusion and exclusion of items and subitems on the RPBS.** Adapted from [24]. RPBS: Rabbit pain behaviour scale. Statistical tests according to Table 3: CV—content validation; % pain $\geq$ 15—at least 15% frequency of occurrence of items/subitems at *pain* time point); PCA—Principal component analysis (loading value $\geq$ 0.50 or $\leq$ -0.50); Intra–intraobserver reliability ($> 0.50$); Inter–inter-observer reliability ($> 0.50$); Resp (responsiveness)—higher score of the behaviour at *pain* time point vs *baseline* according to Friedman test; ITC—item-total Spearman correlation between 0.3–0.7; IC—Internal consistency ($> 0.6$); Sp–Specificity ($\geq$ 70%); S–Sensitivity ($\geq$ 70%). The main items were subjected to ten tests and when approved at least in seven, they were included in the final scale (Table 4); the subitems were subjected to seven tests marked with asterisk (*) and when approved in at least three, they were included in the final scale (Table 4). Number 1 indicates that the item/subitem was approved according to the criteria of each test. The items and subitems included in the final scale after refinement are in bold.
(DOCX)

**S4 Table. Inter-observer matrix agreement of items of the RPBS.** Interpretation of the degree of reliability $k_w$: very good: 0.81–1.0; good: 0.61–0.80; moderate: 0.41–0.60; reasonable: 0.21–0.4; poor $<0.2$ [45]. Bold type corresponds to values $> 0.61$.
(DOCX)

**S1 Video. Moves around normally and/or jumps.**
(MP4)

**S2 Video. Exhibits bipedal position.**
(MP4)

**S3 Video. Exhibits quadrupedal position.**
(MP4)

**S4 Video. Walks at a very slow pace.**
(MP4)

**S5 Video. Lies for most of the time.**
(MP4)

**S6 Video. Does not move for most of the observation time.**
(MP4)

**S7 Video. The rabbit moves normally and/or when stationary performs normal activity.**
(MP4)

**S8 Video. The rabbit moves little and does not perform normal activity.**
(MP4)

**S9 Video. The rabbit is immobile and does not perform normal activity.**
(MP4)

**S10 Video. Interacts with environmental enrichment objects (pinecone).**
(MP4)

**S11 Video. Interacts with environmental enrichment objects (pen substrate).**
(MP4)

**S12 Video. Eats (feed).**
(MP4)

**S13 Video. Eats (carrot).**
(MP4)

**S14 Video. Sniffs the environment.**
(MP4)

**S15 Video. Exhibits body self-cleaning behaviour.**
(MP4)

**S16 Video. Exhibits head self-cleaning behaviour.**
(MP4)

**S17 Video. Keeps eyes wide open.**
(MP4)

**S18 Video. Keeps ears erect.**
(MP4)

**S19 Video. Keeps eyes semi-closed.**
(MP4)

**S20 Video. Keeps eyes closed.**
(MP4)

**S21 Video. Exhibits drooping (semi-lowered) ears.**
(MP4)

**S22 Video. Exhibits drooping (lowered) ears.**
(MP4)

**S23 Video. Licks the affected area.**
(MP4)

**S24 Video. Presses the abdomen against the floor.**
(MP4)

**S25 Video. Keeps one limb suspended.**
(MP4)

**S26 Video. Attempts to stand up, but remains lying down.**
(MP4)

**S27 Video. Rapid dorsal movement of the body (flinches).**
(MP4)

**S28 Video. Retracts and closes the eyes (winces).**
(MP4)

**S29 Video. Tremors.**
(MP4)

**S1 Data.**
(XLSX)

## Acknowledgments

The authors would like to thank Natache Garofalo and Diego Generoso for their contribution in content validation.

## Author Contributions

**Conceptualization:** Renata Haddad Pinho, Stelio Pacca Loureiro Luna, Amy Miller, Paul Flecknell, Matthew C. Leach.

**Data curation:** Renata Haddad Pinho, Pedro Henrique Esteves Trindade, André Augusto Justo, Daniela Santilli Cima, Mariana Werneck Fonseca, Amy Miller, Paul Flecknell, Matthew C. Leach.

**Formal analysis:** Renata Haddad Pinho, Stelio Pacca Loureiro Luna, Pedro Henrique Esteves Trindade, André Augusto Justo, Daniela Santilli Cima, Mariana Werneck Fonseca, Paul Flecknell, Matthew C. Leach.

**Funding acquisition:** Renata Haddad Pinho, Stelio Pacca Loureiro Luna, Bruno Watanabe Minto, Amy Miller, Paul Flecknell, Matthew C. Leach.

**Investigation:** Renata Haddad Pinho, André Augusto Justo, Daniela Santilli Cima, Mariana Werneck Fonseca, Bruno Watanabe Minto, Fabiana Del Lama Rocha, Amy Miller, Paul Flecknell, Matthew C. Leach.

**Methodology:** Renata Haddad Pinho, Stelio Pacca Loureiro Luna, Pedro Henrique Esteves Trindade, André Augusto Justo, Daniela Santilli Cima, Mariana Werneck Fonseca, Bruno Watanabe Minto, Fabiana Del Lama Rocha, Paul Flecknell, Matthew C. Leach.

**Project administration:** Renata Haddad Pinho, Stelio Pacca Loureiro Luna, Bruno Watanabe Minto, Paul Flecknell, Matthew C. Leach.

**Resources:** Stelio Pacca Loureiro Luna, Bruno Watanabe Minto, Fabiana Del Lama Rocha, Paul Flecknell, Matthew C. Leach.

**Software:** Pedro Henrique Esteves Trindade.

**Supervision:** Stelio Pacca Loureiro Luna, Matthew C. Leach.

**Validation:** Renata Haddad Pinho, Stelio Pacca Loureiro Luna, Pedro Henrique Esteves Trindade, Mariana Werneck Fonseca.

**Visualization:** Renata Haddad Pinho, Stelio Pacca Loureiro Luna, André Augusto Justo, Daniela Santilli Cima, Mariana Werneck Fonseca.

**Writing – original draft:** Renata Haddad Pinho.

**Writing – review & editing:** Renata Haddad Pinho, Stelio Pacca Loureiro Luna, Pedro Henrique Esteves Trindade, Matthew C. Leach.

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
