## [Decision Letter · Decision Letter 0]

30 Dec 2021

PONE-D-21-38220Validation of the rabbit pain behaviour scale (RPBS) to assess acute postoperative pain in rabbits (Oryctolagus cuniculus)PLOS ONE

Dear Dr. Luna,

Thank you for submitting your manuscript to PLOS ONE. After careful consideration, we feel that it has merit but does not fully meet PLOS ONE’s publication criteria as it currently stands. Therefore, we invite you to submit a revised version of the manuscript that addresses the points raised during the review process.

We look forward to receiving your revised manuscript.

Kind regards,

Fatih Özden, PhD

Academic Editor

PLOS ONE

Journal Requirements:

Additional Editor Comments:

Please find the reviews of the three reviewers. Pleaser carefully carry out the suggested revisions,

King Regards,

Reviewers' comments:

Reviewer's Responses to Questions

**Comments to the Author**

1. Is the manuscript technically sound, and do the data support the conclusions?

Reviewer #1: Yes

Reviewer #2: Yes

Reviewer #3: Partly

2. Has the statistical analysis been performed appropriately and rigorously? 

Reviewer #1: Yes

Reviewer #2: Yes

Reviewer #3: Yes

3. Have the authors made all data underlying the findings in their manuscript fully available?

Reviewer #1: Yes

Reviewer #2: Yes

Reviewer #3: Yes

4. Is the manuscript presented in an intelligible fashion and written in standard English?

Reviewer #1: Yes

Reviewer #2: Yes

Reviewer #3: No

5. Review Comments to the Author

Reviewer #1: I reviewed the article titled “Validation of the rabbit pain behavior Scale (RPBS) to assess acute postoperative pain in rabbits (Oryctolagus cuniculus)”.

Explanatory and descriptive factor analysis can also offer a different perspective. Or it can be recommended for further studies as a limitation.

Construct validity does not mean responsiveness. It should be corrected.

Do you think the time is appropriate for responsiveness? Provide additional evidence by providing references.

King Regards.

Reviewer #2: Summary

There are very few validated pain scales designed for use in rabbits. This study by Pinho et al. generates and validates a useful novel tool, the rabbit pain behaviour scale (RPBS) for post-operative pain. Pinho et al. demonstrate strong correlations between the RPBS and commonly used unidimensional pain scales (VAS, SDS, NS). A moderate correlation with the Rabbit Grimace Scale is established. Importantly, the authors are the first to establish a threshold at which administration of analgesics is recommended for rabbits. This study provides valuable information to the field, but I do have some concerns and suggest some revisions.

Major Comments

Lines 405-406: “The Youden index determined a score ≥ 3 as the cut-off point to distinguish rabbits in pain from those without pain.”

• Figure 4 shows that rabbits in both ovariohysterectomy groups had median RPBS scores above 5 at baseline/before surgery, which would give the false impression that these rabbits were in pain and in need of analgesics.

• The authors attribute this false positive to methodology (lines 537-546): “Motivational items (pinecones and carrots) were offered at the beginning of each recording only in rabbits undergoing orthopaedic surgery, which stimulated activity… The groups undergoing orchiectomy showed good specificity for the RPBS, even without the presence of stimulating items. This difference in behaviour compared to the OVH groups may due to sex-related behavioural differences, as males are more active than females[57].”

o The use of pinecones and carrots as enrichment prior to filming the rabbits in the Ortho group is very important and should be mentioned more explicitly in the methods section. It’s discussed very well in the discussion.

o What were the sexes of the rabbits in the Ortho group?

o Ref. 57 is not sufficient: “Sex differences in human-directed social behavior in pet rabbits”

o

Lines 485-486: The authors mention that their study was completed prior to the publication of the Bristol Rabbit Pain Scale (BRPS; Benato et al. 2021). It would be valuable to determine the correlation between the RPBS and the BRPS, although this may not be possible. A more detailed discussion of the similarities and differences between the RPBS and the BRPS is warranted.

Lines 518-525: “In this sense, the RPBS showed a high correlation with the unidimensional scales. However, as in the validation of a scale to assess acute pain in sheep [24], the correlation with the facial scale (RbtGS)[18] was only moderate. This result may be related to the lack of training of evaluators to use the RbtGS and/or the lack of refinement of the latter instrument, which did not pass through the important validation criteria such as criterion validity, intra-rater reliability, principal component analysis and internal consistency.”

• It is very concerning that the evaluators were not trained to use the Rabbit Grimace Scale, when a goal of the study was to compare outcomes of the RPBS and the RbtGS. Were the evaluators trained to use the unidimensional scales?

Line 185-187: “The videos of the 58 animals were edited by the main researcher of the study to generate 2 to 3 minute videos that proportionally represented the duration and frequency of the behaviours observed in the original videos.”

• Table 1 shows that rabbits in the ORC-Melox, ORC-Multi, OVH-Pla, and OVH-Melox were originally videotaped for 15-20 minutes. Could you please elaborate on how the videos were shortened to the final length (2-3 minutes)?

Minor Comments

Line 73-86: Please define abbreviations

Table 3 – Typo “IY” instead of “YI”

Line 255 – I suggest adding a brief discussion of the principal component analysis to the discussion section.

Table 7 – The formatting of this table is unclear

Line 604-605 – “The positive side of this bias was the demonstration of the importance of using motivational items in assessments.” Yes, this is great!

Reviewer #3: This work appears to be an extension of the work by Benato et al. ((2021) https://doi.org/10.1371/journal.pone.0252417) and adds validity to RBPS and affirms the practicality of this assessment method in clinical and research settings.

The authors have conducted a comprehensive analysis to conduct their validation. However; there are few points of concern that need to be addressed before such validation is to be accepted and recommended for use in veterinary clinics or animal research laboratories.

Methodology

Evaluation of the videos:

How did the investigator avoid subjective bias while selecting the videos for evaluation? What criteria of inclusion and exclusion were used to minimize bias? It is possible to fall into confirmatory bias and miss essential findings that could have influenced the final conclusions. Please provide detail explanation on this.

Selection and training of evaluators

As in the above comment, there may be potential bias in the process of selection and training. What did you do to avoid/minimize these biases? "All evaluators were veterinarians with residency in veterinary anesthesiology and around 4 years of experience in the area (RHP, AAJ, DSC, and MWF)". It is good that the evaluators have some relevant background; however, such evaluation is better done by professionals with diverse background. Apart from veterinarians, the assessment should be done by animal behaviorists, laboratory animal specialists and researchers with experience in handling and using rabbits. In addition, even the selection of the veterinarian evaluators should include some diversity in terms of training, location and experience (as suggestion, veterinarians from different reputable institutions and veterinary hospitals in different countries).

Conclusion

"The use of motivational items is recommended to ensure an accurate assessment when using the scale." This statement is not clear enough. Please elaborate.

Minor grammatical error corrections. There are few grammatical errors that need to be corrected. For example: Line 133: "...and the anaesthesia maintained with...." Please do through editing and proofreading.

6. PLOS authors have the option to publish the peer review history of their article (what does this mean?). If published, this will include your full peer review and any attached files.

Reviewer #1: No

Reviewer #2: No

Reviewer #3: **Yes: **Erkihun Aklilu

---

## [Author Response · Author response to Decision Letter 0]

7 Apr 2022

PONE-D-21-38220

Validation of the rabbit pain behaviour scale (RPBS) to assess acute postoperative pain in rabbits (Oryctolagus cuniculus)

PLOS ONE

Dear Dr. Luna,

Thank you for submitting your manuscript to PLOS ONE. After careful consideration, we feel that it has merit but does not fully meet PLOS ONE’s publication criteria as it currently stands. Therefore, we invite you to submit a revised version of the manuscript that addresses the points raised during the review process.

We look forward to receiving your revised manuscript.

Kind regards,

Fatih Özden, PhD

Academic Editor

PLOS ONE

Journal Requirements:

Additional Editor Comments:

Please find the reviews of the three reviewers. Pleaser carefully carry out the suggested revisions,

King Regards,

#EDITOR

Dear Editor,

 The authors appreciate the time and effort spent reviewing this manuscript and thank you very much for your comments. All corrections have been performed according to the three Reviewers and each comment responded to separately. 

We hope that after these corrections you consider the manuscript suitable for publication, but we are happy to answer any further questions.

 Yours sincerely,

The authors

#REVIEWER 1

Dear Reviewer,

The authors appreciate the time and effort spent reviewing this manuscript and thank you very much for your comments. All corrections were performed according to your suggestions and each comment was responded separately. 

 We hope that after these corrections you consider the manuscript suitable for publication, but we are happy to answer any further questions.

 Yours sincerely,

The authors

Reviewer #1:I reviewed the article titled “Validation of the rabbit pain behavior Scale (RPBS) to assess acute postoperative pain in rabbits (Oryctolagus cuniculus)”.

Question: Explanatory and descriptive factor analysis can also offer a different perspective. Or it can be recommended for further studies as a limitation.

Answer: Thanks for pointing out this relevant suggestion. After your suggestion we performed Horn's Parallel Analysis (Preacher and MacCallum, 2003)( Marchenko-Pastur limit, and Gavish-Donoho method (Dobriban and Owen, 2019; Gavish and Donoho, 2014) to determine the optimal number of dimensions to be retained. All methods indicated one-dimension, except for the Horn's Parallel Analysis which suggested two-dimensions. In addition, we performed a confirmatory factor analysis (CFA; Bollen 1989) and data suggests that a unidimensional fits better for RPBS. This was included in Methods (page 14, table 3), Results (page 23, Line 268-275) and Discussion (Page 36, Line 555-567). The authors think it is not necessary to include this table in the manuscript, but we are happy to include as a supplemental data if the Reviewer requires.

Question: Construct validity does not mean responsiveness. It should be corrected.

Answer: We apologize for that. The hypothesis testing procedure used for construct validity may be confounded with responsiveness, therefore throughout the manuscript [Methods(Pages 16, table 3), Results ( Page 29, Line 386-398) and Discussion (Page 37, Line 611-615)] corrections have been performed to differentiate construct validity from responsiveness. To avoid any errors, both concepts have been described according to COSMIN checklist, taxonomy and terminology guidelines and another reference (Mokkink et al., 2010; Prinsen et al., 2018; Streiner and Norman, 2016).

Question: Do you think the time is appropriate for responsiveness? Provide additional evidence by providing references

Answer: Yes, this is the current approach reported by the guidelines. Responsiveness refers to the instrument's ability to detect a significant change in a clinical state. We have included references that support the timing of responsiveness assessment after analgesia. (Page 38 – Line 616-624)

Reviewer #2:

Dear Reviewer,

The authors appreciate the time and effort spent reviewing this manuscript and thank you very much for your comments. All corrections were performed according to your suggestions and each comment was responded separately. 

 We hope that after these corrections you consider the manuscript suitable for publication, but we are happy to answer any further questions.

 Yours sincerely,

The authors

Major Comments

Lines 405-406: “The Youden index determined a score ≥ 3 as the cut-off point to distinguish rabbits in pain from those without pain.”

• Figure 4 shows that rabbits in both ovariohysterectomy groups had median RPBS scores above 5 at baseline/before surgery, which would give the false impression that these rabbits were in pain and in need of analgesics.

• The authors attribute this false positive to methodology (lines 537-546): “Motivational items (pinecones and carrots) were offered at the beginning of each recording only in rabbits undergoing orthopaedic surgery, which stimulated activity… The groups undergoing orchiectomy showed good specificity for the RPBS, even without the presence of stimulating items. This difference in behaviour compared to the OVH groups may due to sex-related behavioural differences, as males are more active than females [57].”

Answer: Included as required. Thanks for your comments (Page 38, Line 645-652).

 Question: The use of pinecones and carrots as enrichment prior to filming the rabbits in the Ortho group is very important and should be mentioned more explicitly in the methods section. It’s discussed very well in the discussion.

Answer: Included(Page 11, Line 177-179)

Question: What were the sexes of the rabbits in the Ortho group?

Answer: The ortho group comprised of 11 females and 17 males. This information was presented in table 1 (Page 6), but has also been included in the methods section (Page 10 Line 173-174)

Question:Ref. 57 is not sufficient: “Sex differences in human-directed social behavior in pet rabbits”

Answer: Unfortunately we were not able to find any other reference to further discuss this information. Therefore this sentence was amended. (Page 38, Line 648-652)

Question:Lines 485-486: The authors mention that their study was completed prior to the publication of the Bristol Rabbit Pain Scale (BRPS; Benato et al. 2021). 

It would be valuable to determine the correlation between the RPBS and the BRPS, although this may not be possible. A more detailed discussion of the similarities and differences between the RPBS and the BRPS is warranted.

Answer:Included (Page 35, Line 536-545)

Lines 518-525: “In this sense, the RPBS showed a high correlation with the unidimensional scales. However, as in the validation of a scale to assess acute pain in sheep [24], the correlation with the facial scale (RbtGS)[18] was only moderate. This result may be related to the lack of training of evaluators to use the RbtGS and/or the lack of refinement of the latter instrument, which did not pass through the important validation criteria such as criterion validity, intra-rater reliability, principal component analysis and internal consistency.”

Question: It is very concerning that the evaluators were not trained to use the Rabbit Grimace Scale, when a goal of the study was to compare outcomes of the RPBS and the RbtGS. Were the evaluators trained to use the unidimensional scales?

Answer: The reviewer is right. Considerations regarding these limitations were included (Page 41, Line 728-735).

Line 185-187: “The videos of the 58 animals were edited by the main researcher of the study to generate 2 to 3 minute videos that proportionally represented the duration and frequency of the behaviours observed in the original videos.”

• Table 1 shows that rabbits in the ORC-Melox, ORC-Multi, OVH-Pla, and OVH-Melox were originally videotaped for 15-20 minutes. 

Question:Could you please elaborate on how the videos were shortened to the final length (2-3 minutes)?

Answer: This information was included. (Page 11 Line 191-195)

Minor Comments

Question:Line 73-86: Please define abbreviations

Answer: Included. (Page 4 Line 76-84)

Question:Table 3 – Typo “IY” instead of “YI”

Answer:Corrected. (Page 18, Table 3)

Question:Line 255 – I suggest adding a brief discussion of the principal component analysis to the discussion section.

Answer: Included. (Page 36, Line 555-567)

Question:Table 7 – The formatting of this table is unclear

Answer: Corrected. (Page 26, Table 7)

Question:Line 604-605 – “The positive side of this bias was the demonstration of the importance of using motivational items in assessments.” Yes, this is great!

Answer: Thanks for your comment!

Reviewer #3:

Dear Reviewer,

The authors appreciate the time and effort spent reviewing this manuscript and thank you very much for your comments. All corrections were performed according to your suggestions and each comment was responded separately. 

 We hope that after these corrections you consider the manuscript suitable for publication, but we are happy to answer any further questions.

 Yours sincerely,

The authors

Methodology

Question:Evaluation of the videos:

How did the investigator avoid subjective bias while selecting the videos for evaluation?

What criteria of inclusion and exclusion were used to minimize bias? 

Answer: As all videos were used according to the pre-defined time points, we believe there was no bias in selecting the videos according to the time points, however although the editor of the videos was very careful to include all behaviours present in the unedited videos, we agree that the edition of these videos to shorter periods might have created a bias. In order to minimize that, the behaviours were proportionally selected according to the total period of the videos and this limitation was included (Page 40 Line 701-708). More information was included in methods to better explain how videos were edited as well (Page 11 Line 191-195).

Question: It is possible to fall into confirmatory bias and miss essential findings that could have influenced the final conclusions. Please provide detail explanation on this.

Answer: The information was included in Methods (Page 11, Line 191-195 ) and Limitations (Page 40 Line 701-708). The edition was based on the ethogram recorded in the three previous studies, For example if the rabbit was lying down for 5 minutes during the 15 minute original footage (1/3 of the time), edits were performed to guarantee that the rabbit was lying down for 1 minute of the 3-minute video clip (1 /3 of time).

Selection and training of evaluators

Question: As in the above comment, there may be potential bias in the process of selection and training. What did you do to avoid/minimize these biases?

Answer: This information was included as a limitation (Page 40, Lines 701-708). 

"All evaluators were veterinarians with residency in veterinary anesthesiology and around 4 years of experience in the area (RHP, AAJ, DSC, and MWF)". It is good that the evaluators have some relevant background; however, such evaluation is better done by professionals with diverse background. 

Question: Apart from veterinarians, the assessment should be done by animal behaviorists, laboratory animal specialists and researchers with experience in handling and using rabbits. In addition, even the selection of the veterinarian evaluators should include some diversity in terms of training, location and experience (as suggestion, veterinarians from different reputable institutions and veterinary hospitals in different countries).

Answer: Additional discussion about this topic was included in the limitations. (Page 41, Line 718-724)

Conclusion

Question:"The use of motivational items is recommended to ensure an accurate assessment when using the scale." This statement is not clear enough. Please elaborate.

Answer: Corrected. (Page 42, Line 754-757)

Question:"Minor grammatical error corrections. There are few grammatical errors that need to be corrected. For example: Line 133: "...and the anaesthesia maintained with...." Please do through editing and proofreading.

Answer: Corrected.

---

## [Decision Letter · Decision Letter 1]

12 May 2022

Validation of the rabbit pain behaviour scale (RPBS) to assess acute postoperative pain in rabbits (Oryctolagus cuniculus)

PONE-D-21-38220R1

Dear Dr. Luna,

We’re pleased to inform you that your manuscript has been judged scientifically suitable for publication and will be formally accepted for publication once it meets all outstanding technical requirements.

Kind regards,

Fatih Özden, PhD

Academic Editor

PLOS ONE

Additional Editor Comments (optional):

Reviewers' comments:

Reviewer's Responses to Questions

**Comments to the Author**

1. If the authors have adequately addressed your comments raised in a previous round of review and you feel that this manuscript is now acceptable for publication, you may indicate that here to bypass the “Comments to the Author” section, enter your conflict of interest statement in the “Confidential to Editor” section, and submit your "Accept" recommendation.

Reviewer #1: All comments have been addressed

Reviewer #2: All comments have been addressed

2. Is the manuscript technically sound, and do the data support the conclusions?

Reviewer #1: Yes

Reviewer #2: Yes

3. Has the statistical analysis been performed appropriately and rigorously? 

Reviewer #1: Yes

Reviewer #2: Yes

4. Have the authors made all data underlying the findings in their manuscript fully available?

Reviewer #1: (No Response)

Reviewer #2: Yes

5. Is the manuscript presented in an intelligible fashion and written in standard English?

Reviewer #1: Yes

Reviewer #2: Yes

6. Review Comments to the Author

Reviewer #1: The manuscript should be accepted as it stands. I would like to thank to the authors.

King Regards

Reviewer #2: Pinho et al. have sufficiently addressed all major and minor comments in their revised manuscript and it is now well suited for publication.

7. PLOS authors have the option to publish the peer review history of their article (what does this mean?). If published, this will include your full peer review and any attached files.

Reviewer #1: No

Reviewer #2: No

---

## [Editor Report · Acceptance letter]

17 May 2022

PONE-D-21-38220R1 

Validation of the rabbit pain behaviour scale (RPBS) to assess acute postoperative pain in rabbits *(Oryctolagus cuniculus)*

Dear Dr. Luna:

I'm pleased to inform you that your manuscript has been deemed suitable for publication in PLOS ONE. Congratulations! Your manuscript is now with our production department. 

Kind regards, 

on behalf of

Dr. Fatih Özden 

Academic Editor

PLOS ONE